# Human VPS13A is associated with multiple organelles and influences mitochondrial morphology and lipid droplet motility

Wondwossen M Yeshaw[1], Marianne van der Zwaag[1], Francesco Pinto[1], Liza L Lahaye[1], Anita IE Faber[1], Rubén Gómez-Sánchez[1], Amalia M Dolga[2], Conor Poland[3], Anthony P Monaco[3,4], Sven CD van IJzendoorn[1], Nicola A Grzeschik[1], Antonio Velayos-Baeza[3], Ody CM Sibon[1]*

[1]Department of Cell Biology, University of Groningen, University Medical Center Groningen, Groningen, The Netherlands; [2]Department of Molecular Pharmacology, Groningen Research Institute of Pharmacy (GRIP), Faculty of Science and Engineering, University of Groningen, Groningen, The Netherlands; [3]Wellcome Trust Centre for Human Genetics, Oxford, United Kingdom; [4]Office of the President, Tufts University, Medford, United States

**Abstract** The *VPS13A* gene is associated with the neurodegenerative disorder Chorea Acanthocytosis. It is unknown what the consequences are of impaired function of VPS13A at the subcellular level. We demonstrate that VPS13A is a peripheral membrane protein, associated with mitochondria, the endoplasmic reticulum and lipid droplets. VPS13A is localized at sites where the endoplasmic reticulum and mitochondria are in close contact. VPS13A interacts with the ER residing protein VAP-A via its FFAT domain. Interaction with mitochondria is mediated via its C-terminal domain. In VPS13A-depleted cells, ER-mitochondria contact sites are decreased, mitochondria are fragmented and mitophagy is decreased. VPS13A also localizes to lipid droplets and affects lipid droplet motility. In VPS13A-depleted mammalian cells lipid droplet numbers are increased. Our data, together with recently published data from others, indicate that VPS13A is required for establishing membrane contact sites between various organelles to enable lipid transfer required for mitochondria and lipid droplet related processes.
DOI: https://doi.org/10.7554/eLife.43561.001

**\*For correspondence:**
o.c.m.sibon@umcg.nl

**Competing interests:** The authors declare that no competing interests exist.

## Introduction

The vertebrate VPS13 protein family consists of four closely related proteins, VPS13A, VPS13B, VPS13C and VPS13D (*Velayos-Baeza et al., 2004*). Mutations in *VPS13B, VPS13C* and *VPS13D* are associated with the onset of neurological and developmental disorders (*Kolehmainen et al., 2003*; *Seifert et al., 2009*; *Lesage et al., 2016*; *Gauthier et al., 2018*; *Seong et al., 2018*). Mutations in the *VPS13A* gene are causative for a specific autosomal recessive neurological disorder, Chorea Acanthocytosis (ChAc) (*Rampoldi et al., 2001*; *Ueno et al., 2001*). Most reported *VPS13A* mutations in ChAc patients result in low levels or absence of the protein (*Dobson-Stone et al., 2004*). ChAc patients display gradual onset of hyperkinetic movements and cognitive abnormalities (*Hermann and Walker, 2015*). The function of VPS13A may not be restricted to the brain but also to other tissues since *VPS13A* is ubiquitously expressed in human tissues (*Velayos-Baeza et al., 2004*; *Rampoldi et al., 2001*).

The molecular and cellular function of VPS13 proteins only recently start to emerge. The current knowledge is largely derived from studies about the only *Vps13* gene in *Saccharomyces cerevisiae*. In yeast, Vps13 is a peripheral membrane protein localized at membrane contact sites including nucleus-vacuole, endoplasmic reticulum (ER)-vacuole and endosome-mitochondria contact sites (*Park et al., 2016*; *Lang et al., 2015*; *John Peter et al., 2017*). *Vps13* mutants are synthetically lethal with mutations in genes required to form the ER-mitochondria encounter structure (ERMES) complex (*Park et al., 2016*; *Lang et al., 2015*), suggesting a redundant role of Vps13 at membrane contact sites. In addition, Vps13 is involved in the transport of membrane bound proteins between the trans-Golgi network and prevacuolar compartment (PVC) (*Redding et al., 1996*; *Brickner and Fuller, 1997*) and from endosome to vacuole (*Luo and Chang, 1997*). Vps13 is also required for pro-spore expansion, cytokinesis, mitochondria integrity, membrane contacts and homotypic fusion and the influential role of Vps13 in these processes is postulated to be dependent on the availability of phosphatidylinositides (*Park et al., 2016*; *Lang et al., 2015*; *John Peter et al., 2017*; *Park and Neiman, 2012*; *Nakanishi et al., 2007*; *De et al., 2017*; *Rzepnikowska et al., 2017*).

The *VPS13A* gene is located at chromosome 9q21 and encodes a high molecular weight protein of 3174 amino acids (*Velayos-Baeza et al., 2004*; *Rampoldi et al., 2001*; *Ueno et al., 2001*). In various model systems, loss of VPS13A is associated with diverse phenotypes, such as impaired autophagic degradation, defective protein homeostasis (*Muñoz-Braceras et al., 2015*; *Lupo et al., 2016*; *Vonk et al., 2017*), delayed endocytic and phagocytic processing (*Korolchuk et al., 2007*; *Samaranayake et al., 2011*), actin polymerization defects (*Föller et al., 2012*; *Alesutan et al., 2013*; *Schmidt et al., 2013*; *Honisch et al., 2015*) and abnormal calcium homeostasis (*Yu et al., 2016*; *Pelzl et al., 2017*). Proteomic studies revealed that VPS13A is associated with multiple cellular organelles (*Huttlin et al., 2015*; *Zhang et al., 2011*; *Hung et al., 2017*) suggesting that VPS13A probably plays a role in a multitude of cellular functions and its loss of function could be associated with a wide range of cellular defects in eukaryotes. Here, to understand the versatile role of VPS13A at the molecular level, the subcellular localization, binding partners and the role of the domains of VPS13A were studied in mammalian cells. We used biochemical and sub-cellular localization studies and demonstrated that VPS13A is associated to multiple cellular organelles including at areas where mitochondria and ER are in close proximity and at lipid droplets. By using CRISPR/Cas9 a *VPS13A* knock-out cell-line was generated to investigate these organelles under VPS13A-depleted conditions. Part of the observed phenotype is also present in a *Drosophila melanogaster Vps13* mutant, a phenotype rescued by overexpression of human VPS13A in the mutant background, indicating a conserved function of this protein. We discuss how our findings, in combination with other recently published VPS13A-related manuscripts, are consistent with an ERMES-like role for VPS13A at membrane contact sites in mammalian cells.

## Results

### Human VPS13A is a peripheral membrane protein

To determine the subcellular localization of endogenous human VPS13A, we first used a biochemical approach and the membrane and cytosolic fractions of HeLa cells were separated by high-speed centrifugation. VPS13A was enriched in the pellet, which contained the transmembrane epidermal growth factor receptor (EGFR) and relatively little of α-tubulin, a cytosolic marker protein (*Figure 1A*, *Figure 1—figure supplement 1*). To further investigate the membrane association of VPS13A, a detergent based subcellular fractionation was performed in HEK293T cells (*Holden and Horton, 2009*). Following digitonin treatment and centrifugation, more than 80% of VPS13A, remained in the fraction containing membrane associated proteins such as EGFR and the ER integral protein- VAMP-associated protein A (VAP-A), and little VPS13A was detected in the cytosolic non-membrane bound and GAPDH containing fraction (*Figure 1B and B'*). The type of membrane association of VPS13A was further investigated by assessing its dissociation from lipid bilayers after treatment with different chemical agents. Similarly to ATP5A, a peripheral membrane associated protein of mitochondria, part of VPS13A was solubilized by alkaline and urea-containing solutions. In contrast, the integral membrane protein EGFR was not solubilized by alkaline containing solutions and was, as expected, only partly removed by urea containing solutions (*Figure 1C,C'*). Altogether, these analyses suggest that VPS13A is a peripheral membrane-associated protein.

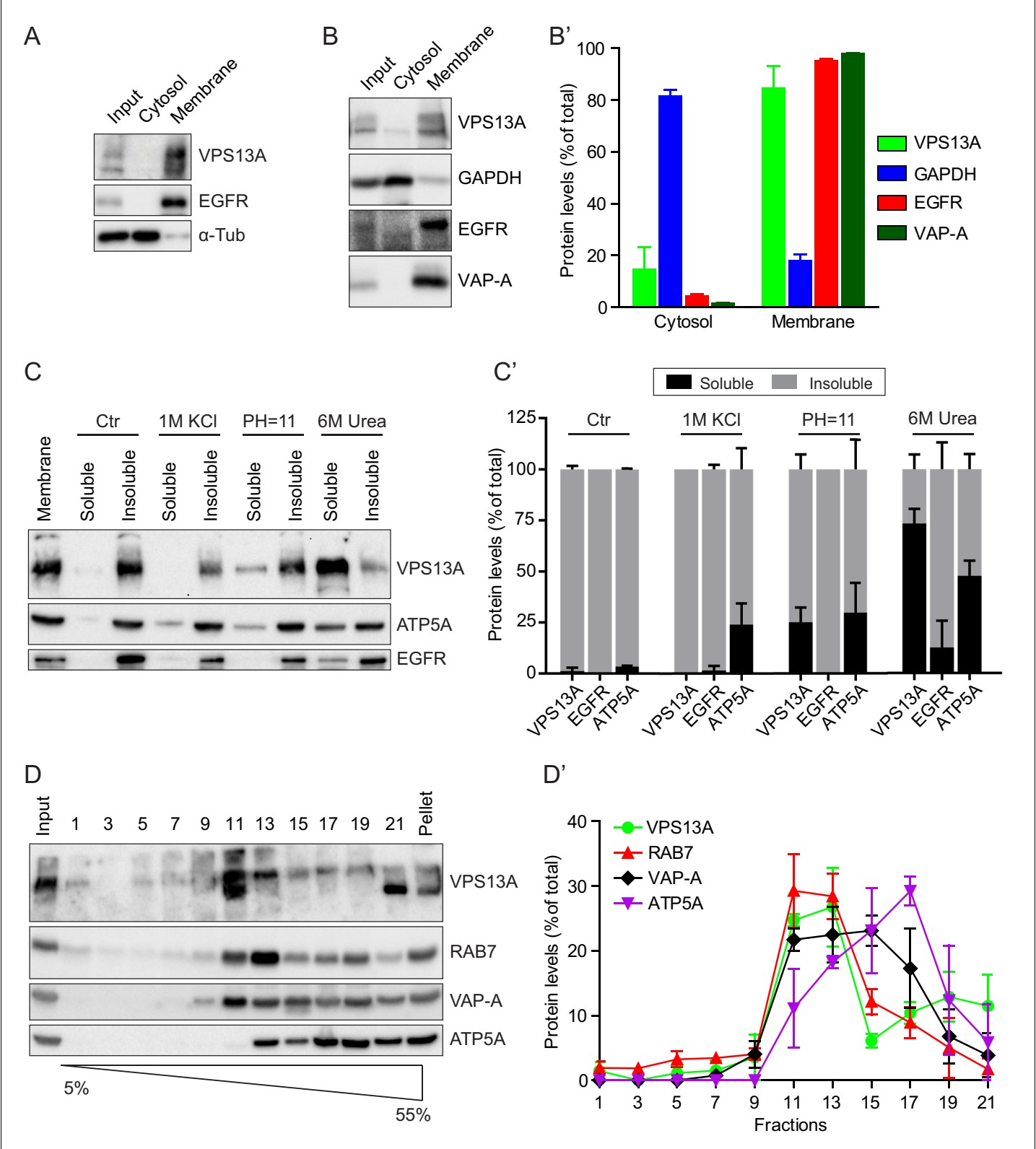

**Figure 1.** VPS13A is enriched in membrane fractions and is peripherally associated to membranes. (**A**) Light membrane fractions from HeLa cell homogenates were separated by centrifugation in a cytosolic and a membrane fraction. Equal amounts of proteins were processed for immunoblot analysis of VPS13A, EGFR and α-tubulin. (**B**) Digitonin extraction of cytosolic proteins in HEK293T cells were immunoblotted for the indicated proteins. The amount of protein was quantified using ImageJ and presented as a percentage of the total (**B'**). (**C**) Membrane fractions of HeLa cells were

*Figure 1 continued on next page*

*Figure 1 continued*

prepared as in A and subjected to different chemical agents to extract proteins from membranes. Equal amount of proteins were processed for immunoblotting using antibodies against VPS13A, EGFR and ATP5A. The amount of protein was quantified using ImageJ and presented as a percentage of the total (**C'**) (**D**) Sucrose gradient fractionation from HeLa cells. HeLa cells were lysed in detergent free buffer and separated in 5–55% sucrose gradients by high speed centrifugation. After TCA precipitation, fractions were processed for immunoblotting using antibodies against VPS13A, VAP-A, RAB7 and ATP5A. Quantification of protein band intensities in D was performed using ImageJ and plotted as percentage of the total (**D'**). In B', C', D', error bars, mean ±s.e.m (n = 3).

DOI: https://doi.org/10.7554/eLife.43561.002

The following figure supplement is available for figure 1:

**Figure supplement 1.** Scan of original blots for Figure 1.

DOI: https://doi.org/10.7554/eLife.43561.003

To determine to which intracellular membranes endogenous VPS13A is associated, we performed subcellular fractionation experiments on a sucrose gradient. These experiments showed that VPS13A was predominantly detected in fractions containing VAP-A, Rab7 and ATP5A, which are marker proteins of the ER, endosomes and mitochondria respectively (*Figure 1D,D'*).

## VPS13A localization to mitochondria is mediated via the C-terminal end

To characterize the subcellular localization of VPS13A in more detail, GFP- and Myc-tagged VPS13A were expressed in HEK293T cells. This yielded a high molecular weight band, corresponding to full-length tagged VPS13A (*Figure 2—figure supplement 1*). Under normal growth conditions, VPS13A-GFP showed two main subcellular distribution patterns. In most cells, VPS13A-positive filamentous structures (*Figure 2A,A'*) and/or punctated or vesicular-like structures (*Figure 2B', B'*) were observed. To identify these compartments, we co-localized VPS13A with a variety of organelle marker proteins. Although not co-localizing with the endosomal/lysosomal marker proteins Rab5, Rab7, LAMP1 and FYCO1 (*Figure 2—figure supplements 1–2*), VPS13A-GFP strongly decorated the periphery of nearly all mitochondria stained with Mitotracker (*Figure 2C,C', C"* and *Video 1*).

To determine whether endogenous VPS13A is a mitochondrial membrane protein, crude mitochondria fractions isolated by centrifugation were analyzed by immunoblotting. VPS13A was highly enriched in the mitochondria fraction and slightly in the microsomal (pellet) fraction (*Figure 3—figure supplements 1–2*). For the alkaline treatment, crude mitochondria fractions were incubated with 0.1 M $Na_2CO_3$ (pH = 11.5). In this experiment, TOMM20 and ATP5A, which are integral and peripheral mitochondria membrane proteins respectively, served as markers. While TOMM20 was mostly retained in the insoluble membrane fraction following $Na_2CO_3$ treatment, VPS13A was now also found in the soluble supernatant in a similar way as ATP5A (*Figure 3—figure supplements 1–2*). Moreover, when crude mitochondria fractions were treated with proteinase K (PK), both TOMM20 and VPS13A were stripped off, suggesting that VPS13A is exposed to the cytosol (*Figure 3—figure supplements 1–2*).

This interesting VPS13A localization to the mitochondria surface prompted us to determine the VPS13A domain that mediates this localization. To do so, GFP-tagged truncated forms of VPS13A (*Figure 2D* and *Figure 3—figure supplements 3–4*) were expressed in U2OS cells, which are more stretched out and possess less rounded and more elongated mitochondria, as compared to HEK293T, and would therefore be better suitable for these imaging studies. Most of these constructs showed an apparently cytosolic distribution pattern except, the C-terminal region of VPS13A (aa 2615–3174) which showed a localization pattern similar to that of the mitochondrial outer membrane marker TOMM20 (*Figure 2E*). Note that, although mitochondria of U2OS possess a different shape, compared to HEK293T cells, VPS13A (aa 2615–3174) localizes in both cell lines in close vicinity to mitochondria (*Figure 3—figure supplement 3*) Analysis of co-localization studies using Mitotracker and VPS13A (aa 2615–3174) showed that the VPS13A signal is localized at the periphery rather than within mitochondria (*Figure 3—figure supplement 3*). This strongly suggests that the C-terminal region of VPS13A is involved in targeting the protein to close vicinity of the outer mitochondrial membrane.

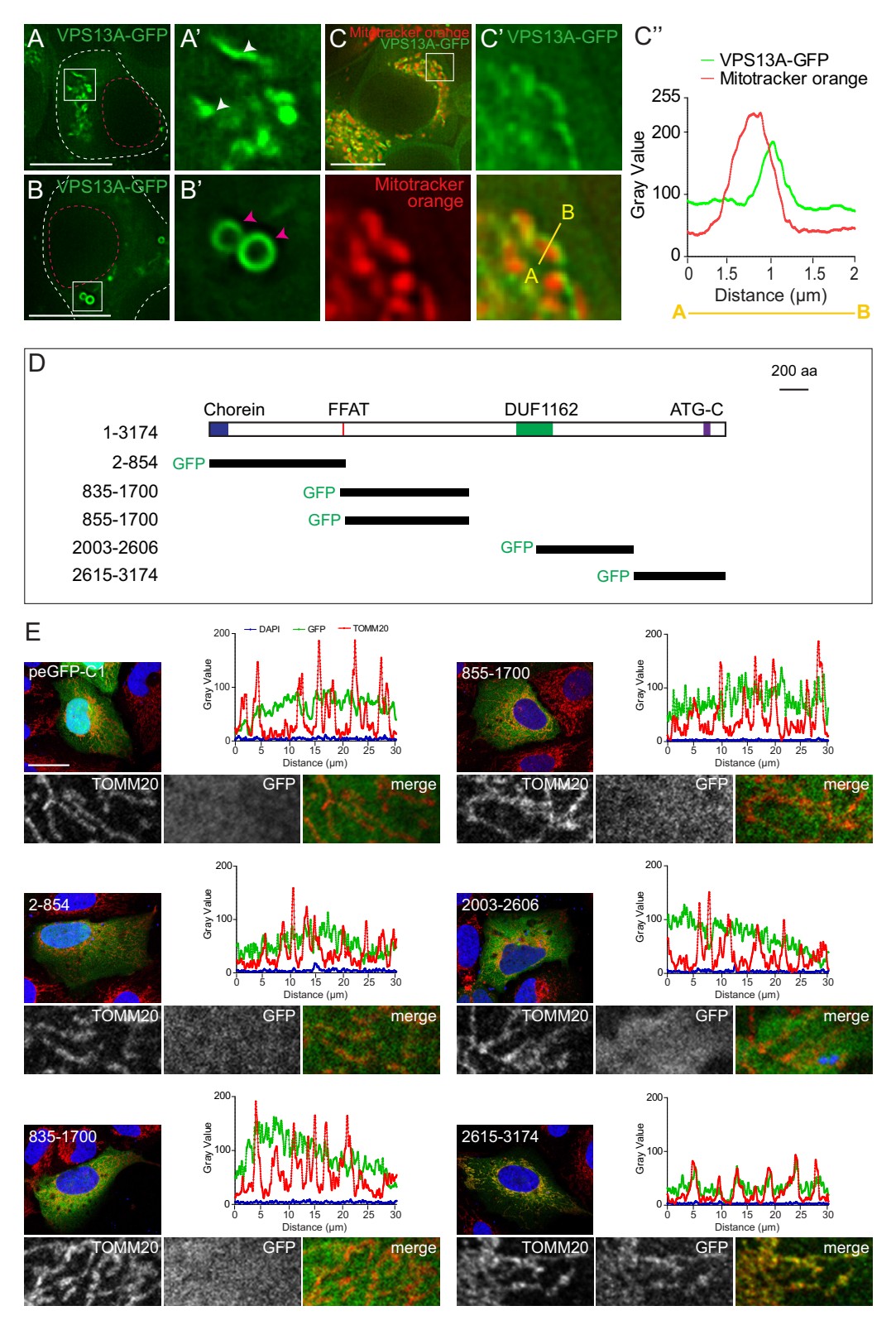

**Figure 2.** VPS13A is localized at mitochondria via its C-terminal domain. (**A,B**) HEK293T cells were transfected with VPS13A-GFP and the GFP signal was visualized using confocal microscopy. White arrowheads show reticular structures (**A, A'**) and magenta arrowheads show vesicular structures (**B, B'**). Cell borders are marked by white dashed lines and the nucleus is marked by magenta dashed lines. (**C**) Single stack image from a time-lapse recording of HEK293T cells expressing VPS13A-GFP for 48 hr (*Video 1*). Mitochondria were labeled using Mitotracker orange. C', C' Line scan analysis of VPS13A-
*Figure 2 continued on next page*

*Figure 2 continued*
GFP and Mitotracker orange indicates the peri-mitochondrial localization of VPS13A. (D) Schematic representations of full length VPS13A and N-terminally GFP tagged VPS13A fragments. Numbers denote the first and last amino acid positions. (E) GFP-VPS13A (green) constructs represented in D were overexpressed in U2OS cells for 24 hr. Cells were stained for TOMM20 (red) and DAPI (blue). Line scan co-localization analysis was done for all channels. Scale bars = 10 μm (**A–C**) and 25 μm (**E**).
DOI: https://doi.org/10.7554/eLife.43561.004
The following figure supplements are available for figure 2:
**Figure supplement 1.** VPS13A colocalizes with mitochondria but not with the endocytic compartment.
DOI: https://doi.org/10.7554/eLife.43561.005
**Figure supplement 2.** Scan of original blots for *Figure 2—figure supplement 1*.
DOI: https://doi.org/10.7554/eLife.43561.006

## VPS13A localizes to the ER-mitochondria interface

Furthermore, the VPS13A localization pattern partly overlapped with the ER markers VAP-A and BFP-Sec61B (yellow signal in *Figure 3A*, white arrowheads in *Figure 3B,C*). Note that in areas where VPS13A and Sec61B or VAP-A are in close contact, a Mitotracker or TOMM20-positive signal is present as well (white arrowheads in *Figure 3B,C*), in contrast to locations positive for an ER marker and negative for VPS13A (magenta arrows in *Figure 3B*). To further investigate the localization pattern of VPS13A in relation to the ER, we conducted time-lapse imaging of live cells expressing VPS13A-GFP and mCherry-VAP-A. This analysis showed that VPS13A-GFP was closely associated to VAP-A positive regions of the ER, the signals partially overlapped, and the dynamics of the VPS13A positive regions are similar to the ER dynamics (*Figure 3D* and *Video 2*). Given the peripheral-membrane protein characteristics of VPS13A, the decoration of mitochondria with VPS13A-GFP, its enrichment in the outer mitochondria membrane and its close association with VAP-A positive ER regions, these results suggest that VPS13A was enriched at the interface between these two organelles, rather than being localized in the interior of both mitochondria and ER.

## VPS13A directly binds VAP-A through its FFAT motif

We then asked what mediated the VPS13A association to the ER. Several membrane-associated proteins bind to the ER resident protein VAP-A through a seven amino acids FFAT motif (*Loewen et al., 2003*; *Loewen and Levine, 2005*; *Murphy and Levine, 2016*). Interestingly, VPS13A also contains a putative FFAT motif (*Murphy and Levine, 2016*), which is located between amino acids 842–848 (*Figure 4A*). To test whether VPS13A indeed interacts with VAP-A, we performed co-immunoprecipitation experiments with endogenous proteins. In line with this hypothesis, VAP-A was enriched in immunoprecipitates of endogenous VPS13A (*Figure 4B*, *Figure 4—figure supplements 1–2*). Conversely, VPS13A was present in the VAP-A immunoprecipitates (*Figure 4B'*).

To test whether VPS13A and VAP-A interact via the putative VPS13A FFAT motif, we conducted a set of in vitro pull-down experiments. We generated GST-tagged recombinant VPS13A fragments (*Figure 4C*) that were incubated with bacterially expressed 6x-His tagged VAP-A. We found that all the constructs containing the VPS13A FFAT motif were efficiently binding VAP-A (*Figure 4D*), including the FFAT motif itself (*Figure 4D*, Lane 3). Importantly, the introduction of the D845A point mutation in this motif, which is known to affect VAP-A binding in other FFAT-containing proteins (*Loewen et al., 2003*; *Saita et al., 2009*), reduced its association to VAP-A (*Figure 4D*, Lane 6). Similar results were obtained when these GST-tagged recombinant VPS13A fragments were incubated with HeLa cell lysates. Following GST pull down, endogenous VAP-A from HeLa cells was found to be enriched together with GST-VPS13A fragments

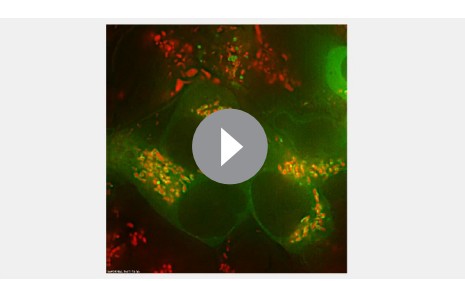

**Video 1.** HEK 293 T cells overexpressing VPS13-GFP were incubated with mitotracker orange for 20 min. Time lapse images were taken every 500 milliseconds and the video is played at 10 frames per second.
DOI: https://doi.org/10.7554/eLife.43561.007

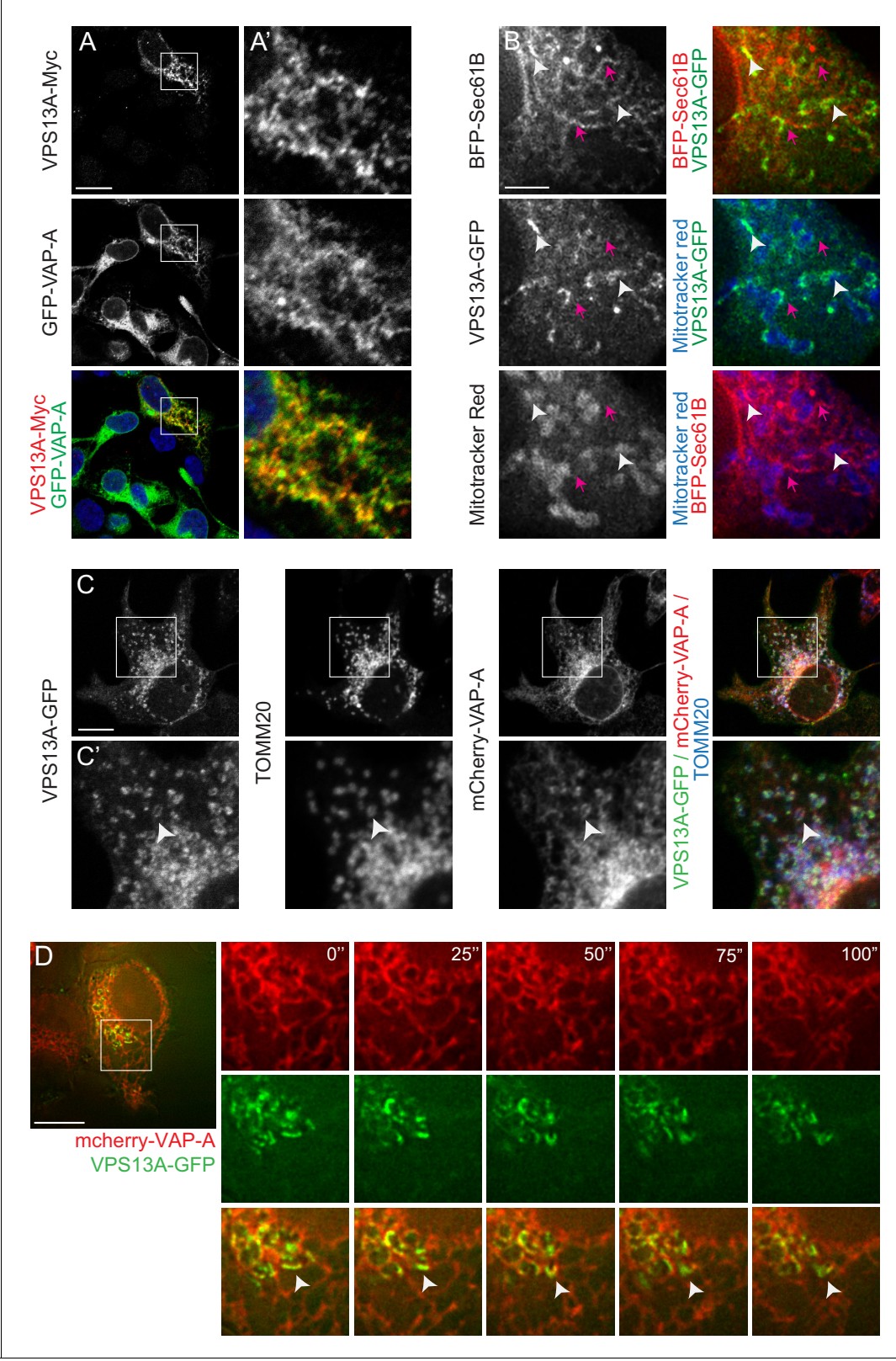

**Figure 3.** VPS13A is localized at the ER-mitochondria interface. (**A**) HEK293T cells were co-transfected with VPS13A-Myc and the ER marker GFP-VAP-A. Cells were stained with anti-Myc (red) and DAPI (blue). A' shows higher magnification of the inserts in A. (**B**) Representative single stack image of HEK293T cells expressing the ER marker BFP-Sec61B and VPS13A-GFP. Mitochondria were labeled using Mitotracker red. White arrowheads indicate the enrichment of VPS13A at the ER-mitochondria interface. Magenta arrows indicate BFP-Sec61B positive ER tubules, negative for VPS13A-GFP and
*Figure 3 continued on next page*

*Figure 3 continued*

not in close association with mitochondria. (**C**) Representative single stack image of HEK293T cells expressing mCherry-VAP-A (ER marker) and VPS13A-GFP. Mitochondria were labeled using TOMM20 antibody. White arrowheads indicate the enrichment of VPS13A at areas positive for ER and mitochondria markers. C' shows higher magnification of the insert in C. (**D**) Representative time-lapse images of HEK293T cells expressing VPS13A-GFP and mCherry-VAP-A for 48 hr (*Video 2*). White arrowheads points to continuous dynamic associations of VPS13A-GFP and mCherry VAP-A. Scale bars = 10 μm (**A, C, D**), and 2 μm (**B**).

DOI: https://doi.org/10.7554/eLife.43561.008

The following figure supplements are available for figure 3:

**Figure supplement 1.** VPS13A is enriched in fractions of the outer mitochondria membrane.
DOI: https://doi.org/10.7554/eLife.43561.009

**Figure supplement 2.** Scan of original blots for *Figure 3—figure supplement 1*.
DOI: https://doi.org/10.7554/eLife.43561.010

**Figure supplement 3.** VPS13A interacts with VAP-A in human cells.
DOI: https://doi.org/10.7554/eLife.43561.011

**Figure supplement 4.** Scan of original blots for *Figure 3—figure supplement 3*.
DOI: https://doi.org/10.7554/eLife.43561.012

in a FFAT-dependent manner (*Figure 4—figure supplements 3–4*). These results indicate that VPS13A interacts with VAP-A via its FFAT domain.

To investigate whether the FFAT motif is required for the localization of VPS13A to the ER, we generated a VPS13A FFAT-deletion mutant (VPS13A$^{\Delta FFAT}$) tagged with GFP. Analysis of confocal images showed that VPS13A$^{\Delta FFAT}$ still presented co-localization to mitochondria comparable to the full length (*Figure 4—figure supplement 3*, yellow signal in the overlay images) but no co-localization was observed between ER-marker VAP-A and VPS13A$^{\Delta FFAT}$ (absence of yellow signal in the overlay image of VPS13A$^{\Delta FFAT}$ and VAP-A, (*Figure 4E'*, *Figure 4—figure supplement 3*), indicating that the FFAT domain is the main hub for ER targeting of VPS13A. The FFAT domain appeared not to be sufficient for an in vivo association with the ER, since FFAT containing VPS13A fragments appeared to remain cytosolic and did not show a reticular pattern (*Figure 2D,E*). To further investigate the requirement of the FFAT domain in the interaction with VAP-A, we expressed VPS13A-GFP$^{\Delta FFAT}$ and found no immunoprecipitation with endogenous VAP-A, whereas the full length construct did (*Figure 4F*).

The assembly of membrane contact sites is regulated by cellular calcium levels (*Giordano et al., 2013*; *Idevall-Hagren et al., 2015*). Calcium levels are mainly regulated through the activity of sarcoendoplasmic reticulum calcium ATPase (SERCA), which can be pharmacologically inhibited with thapsigargin (TG), leading to an increase in cytosolic calcium. In order to understand the effect of cellular calcium on VPS13A-VAP-A interaction, we treated cells with different concentrations of TG. GFP-VAP-A was expressed in HeLa cells and after TG treatment GFP-trap assays were used to immunoprecipitate GFP-VAP-A and an increased amount of endogenous VPS13A bound to GFP-VAP-A was observed (*Figure 4G,H*). The increase was proportional to the concentration of TG applied. The calcium mediated VPS13A-VAP-A interaction suggests that VPS13A plays a role in ER-mitochondria contact sites.

In conclusion, our data support a model where VPS13A can associate simultaneously with mitochondria and ER via its C-terminus and FFAT domain, respectively.

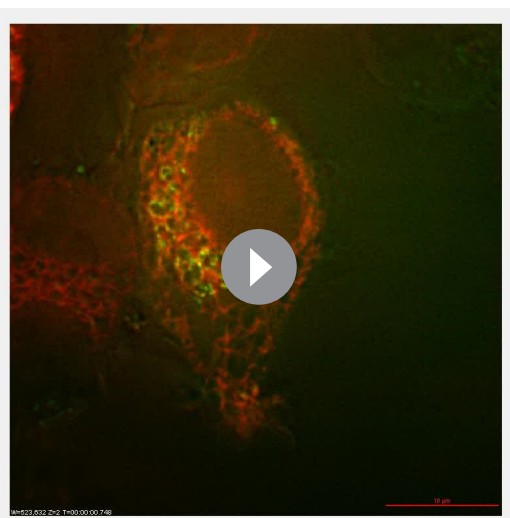

**Video 2.** HEK 293 T cells overexpressing VPS13-GFP and mCherry-VAP-A were imaged lapse images were taken every 5 s. The video is played at five frames per second.
DOI: https://doi.org/10.7554/eLife.43561.013

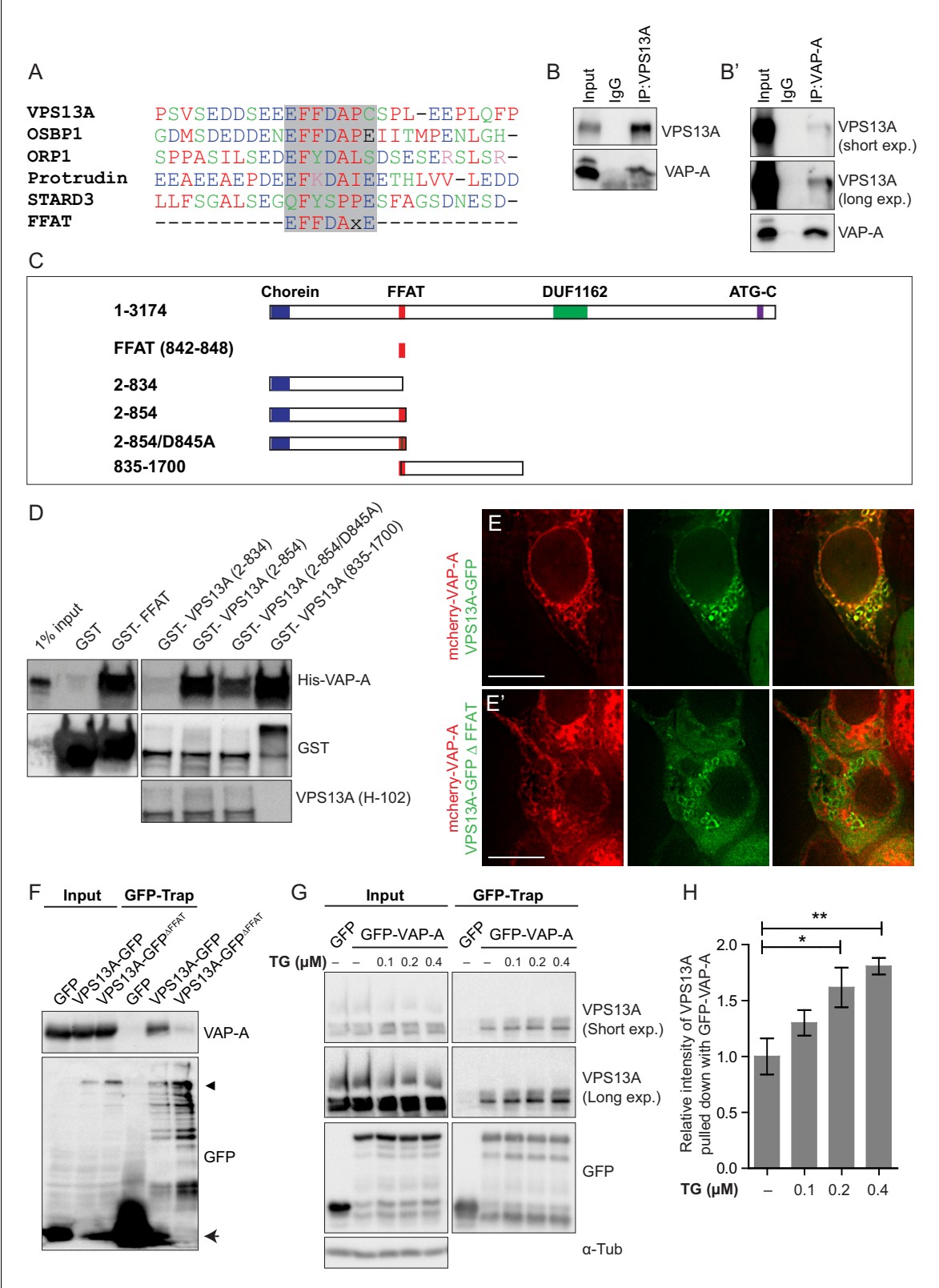

**Figure 4.** Direct interaction of VPS13A and VAP-A. (**A**) Amino acid sequence alignment of VPS13A-FFAT and four other FFAT containing proteins. The FFAT containing region (gray box) of each protein was selected and aligned using ClustalW multiple alignment tool. (**B**) Endogenous VPS13A was immunoprecipitated from HeLa cells using an anti-VPS13A antibody. Rabbit IgG was used as a control. (**B'**) Endogenous VAP-A was immunoprecipitated from HeLa cells using an anti-VAP-A antibody. Goat IgG was used as a control. Indicated proteins were detected by

*Figure 4 continued on next page*

*Figure 4 continued*

immunoblotting. (C) Schematic representations of bacterially expressed GST tagged VPS13A fragments used for the in vitro binding assays in D. (D) In vitro binding assay using 6xHis-VAP-A and GST-fusions of VPS13 fragments (depicted in C) expressed in *E.Coli*. GST-fusion proteins were enriched on Sepharose beads and incubated with equal amounts of bacterial lysate containing 6xHis-VAP-A. GST alone used as a control. Samples were immunoblotted against VAP-A, GST and N-terminal VPS13A (H-102). (E) Representative single stack image of HEK293T cells expressing mCherry-VAP-A (red) and VPS13A-GFP (E) or VPS13A-GFP $^{\Delta FFAT}$ (E'). A yellow signal in the overlay indicates a close association between VPS13A-GFP and VAP-A (E) and the absence of a yellow signal indicates the absence of a close association between VPS13A-GFP $^{\Delta FFAT}$ and VAP-A (E'). (F) GFP tagged full length VPS13A and VPS13A $^{\Delta FFAT}$ were transiently expressed in HEK293T cells. Cell lysates were immunoprecipitated using a GFP-trap assay. GFP alone was used as a control. Indicated proteins were detected by immunoblotting. Arrowhead indicates the VPS13A-GFP band and arrow indicates free GFP band. (G) GFP-VAP-A was immunoprecipitated from HeLa cells treated with different concentrations of Thapsigargin (TG) for 6 hr. DMSO was used as control. Indicated proteins were detected by immunoblotting. (H) Densitometric quantification of protein bands in G. The ratio of immunoprecipitated VPS13A was normalized to the respective amount of GFP-VAP-A. Cells treated with DMSO were used as controls. Data above (B, D, F) represents (n = 3), in H, error bars, mean ±s.e.m (n = 3), two-tailed unpaired Student's t-test was used (*p≤0.05, **p≤0.01). Scale bars = 10 μm (E, E').
DOI: https://doi.org/10.7554/eLife.43561.014

The following figure supplements are available for figure 4:

**Figure supplement 1.** Scan of original blots for *Figure 4*.
DOI: https://doi.org/10.7554/eLife.43561.015

**Figure supplement 2.** Scan of original blots for *Figure 4*.
DOI: https://doi.org/10.7554/eLife.43561.016

**Figure supplement 3.** VPS13A interacts with VAP-A.
DOI: https://doi.org/10.7554/eLife.43561.017

**Figure supplement 4.** Scan of original blots for *Figure 4—figure supplement 3*.
DOI: https://doi.org/10.7554/eLife.43561.018

## Depletion of VPS13A is associated with decreased areas of proximity between ER and mitochondria

Our results so far indicate that VPS13A is localized, among others, at areas were the ER and mitochondria are in close proximity. We aimed to investigate a possible role for VPS13A in influencing ER-mitochondria contact sites. We used a split-GFP-based contact site sensor (SPLICS) engineered to fluoresce when organelles are in proximity (*Cieri et al., 2018*). This assay consists of co-expression of two constructs, one encoding a non-fluorescent portion of GFP fused to an ER-targeting signal, and another one encoding a complementing non-fluorescent portion of GFP fused to an OMM moiety targeting it to the cytoplasmic side of the outer mitochondrial membrane. When in close contact, the two non-fluorescent portions of GFP fold and a fluorescent GFP is obtained. We used two variants, named SPLICS$_S$ and SPLICS$_L$, detecting narrow ($\approx$8–10 nm) and wide ($\approx$40–50 nm) distances between ER and mitochondria respectively. Contact sites between ER and mitochondria result in bright spots (*Cieri et al., 2018*). In order to investigate a possible role of VPS13A in ER-mitochondria contact sites, we used a MCR5 *VPS13A* KO cell line, obtained via a CRISPR/Cas9 approach, with no detectable levels of VPS13A protein while its closest homologous protein VPS13C appears normal (*Figure 5—figure supplements 1–2*). In these cells using the SPLICS sensor, contact sites could be visualized (*Figure 5A,B*) as previously reported (*Cieri et al., 2018*). Both signals from the SPLICS assay, for narrow and wide distances, are significantly decreased in VPS13A depleted cells compared to the parental cell line (*Figure 5A''–B''*). Together our results not only indicate that VPS13A is present at areas were mitochondria and ER are in close proximity, but also that VPS13A is involved in the formation or stabilization of ER-mitochondria contact sites.

## Mitochondria elongation is impaired in VPS13A depleted cells

ER-mitochondria contact sites are required for the transfer of lipids between the ER (where majority of lipid synthesis occurs) and mitochondria (*Gatta and Levine, 2017*) and, therefore, a decrease in ER-mitochondria contact sites may have consequences for mitochondria processes such as fission, fusion and mitophagy which are all influenced by the lipid composition of mitochondria membranes (*Böckler and Westermann, 2014*; *Lahiri et al., 2015*). We used the VPS13A KO cell line to investigate the consequences of VPS13A depletion in these processes. Upon morphological examination we found that VPS13A depleted cells contained less elongated mitochondria compared to control cells when cultured under standard conditions (*Figure 5C,C'*).Upon starvation, a process which

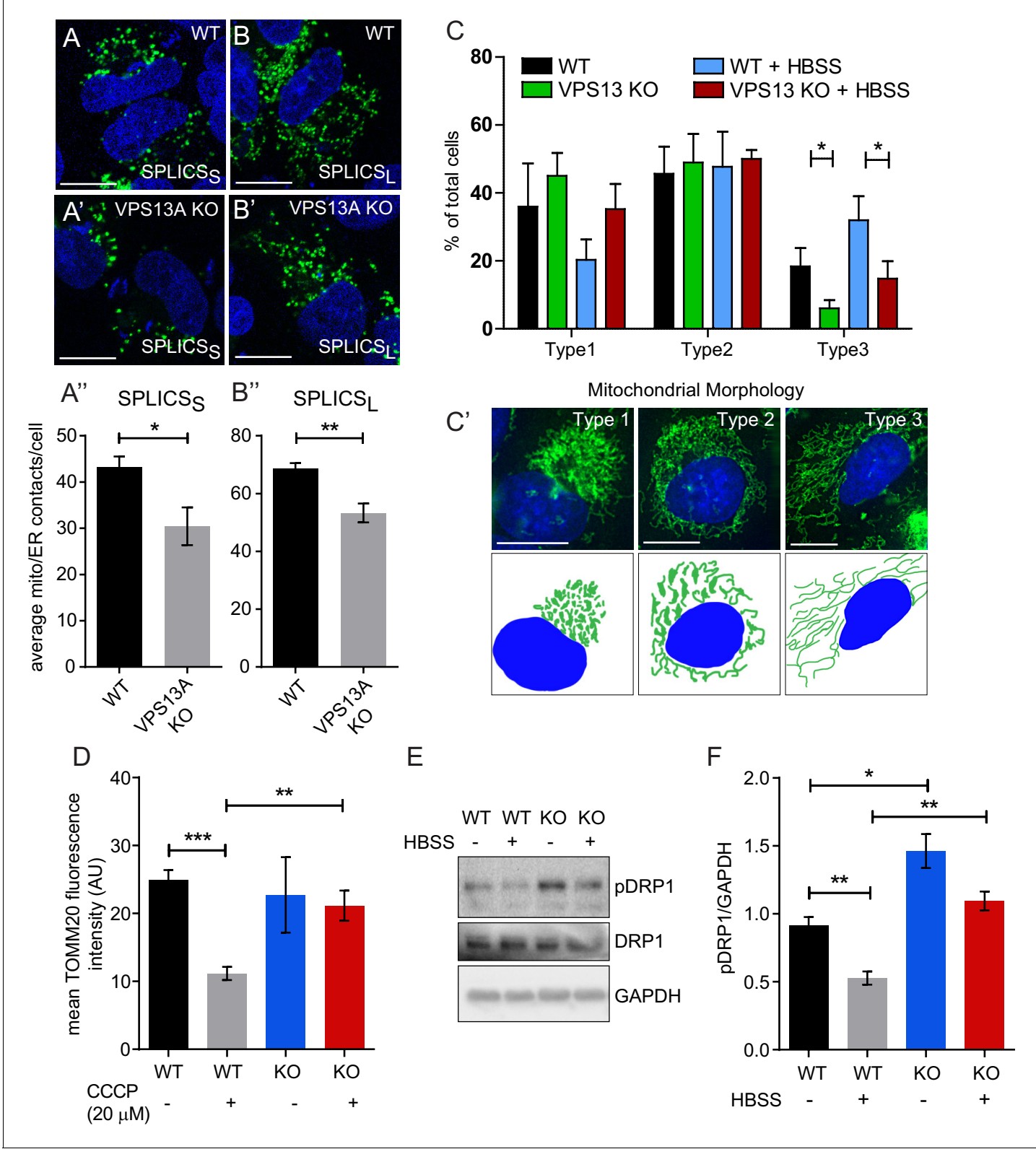

**Figure 5.** Depletetion of VPS13A results in less elongated mitochondria and impaired mitophagy. (A–B) Representative images of control MRC5 cells (WT) (**A, B**) and VPS13 KO MRC5 cells (**A', B'**) transfected with SPLICS$_S$ (**A, A'**) or SPLICS$_L$ (**B, B'**) to detect narrow (~8–10 nm) or long (~40–50 nm) distance ER-mitochondria contact sites respectively. A', B'. Quantification of narrow and long distance contact sites in WT and VPS13A KO MRC5 cells. Error bars, mean ±s.e.m (n = 3 (**A''**) and n = 5 (**B''**)), two-tailed unpaired Student's t-test was used (*p≤0.05, **p≤0.01). (**C**) Quantification of the
*Figure 5 continued on next page*

*Figure 5 continued*

mitochondria morphology of cells cultured under normal conditions (control) or under starved conditions (HBSS). WT and VPS13A KO MRC5 cells, were stained for the mitochondria marker TOMM20 (red) and DAPI (blue). For the quantification three cell-types with different mitochondrial appearances were pre-defined, type 1 cells with short, fragmented, densely packed mitochondria, type 2 cells with a mixture of round densely packed and more tubulated and less densely packed mitochondria and type 3 cells with tubulated dispersed and long mitochondria. Typical images and schematics are provided(C'). Error bars, mean ±s.e.m (n = 3), two-tailed unpaired Student's t-test was used (*p≤0.05). (D) Mitophagy assay of control MRC5 (WT) and VPSA13 KO cells. The cells were transfected with FLAG-Parkin, which allows for the removal of damaged mitochondria and were treated with DMSO (control) or 20 μM CCCP (inducing mitochondria damage). After the transfection/treatment the cells were stained for the mitochondria marker TOMM20 and the mean fluorescence TOMM20 intensity was measured exclusively in FLAG-Parkin positive cells. The decrease in TOMM20 fluorescence after CCCP represents mitophagy. Error bars, mean ±s.e.m (n = 3), two-tailed unpaired Student's t-test was used (**p≤0.01, ***p≤0.001). (E,F) In control and starved (HBSS) MRC5 WT and VPS13A KO cells levels of pDRP1 and total DRP1 were determined by immunoblotting using GAPDH as a loading control (E). Quantification of protein band intensities in F was performed using ImageJ and plotted as a ratio of pDRP to GAPDH (F). Error bars, mean ±s.e.m (n = 3), two-tailed unpaired Student's t-test was used (*p≤0.05, **p≤0.01, ***p≤0.001). Scale bars = 10 μm (A, A', B, B', C').

DOI: https://doi.org/10.7554/eLife.43561.019

The following figure supplements are available for figure 5:

**Figure supplement 1.** Validation of the VPS13A mutant cell line (VPS13A KO).

DOI: https://doi.org/10.7554/eLife.43561.020

**Figure supplement 2.** Scan of original blots for *Figure 5—figure supplement 1*.

DOI: https://doi.org/10.7554/eLife.43561.021

**Figure supplement 3.** Scan of original blots for *Figure 5*.

DOI: https://doi.org/10.7554/eLife.43561.022

induces the formation of elongated mitochondria (*Rambold et al., 2011*), an increased amount of VPS13A KO cells with elongated mitochondria was observed, however, not to the extent as observed in control cells (*Figure 5C,C'*). Finally, a reduced capacity to eliminate damaged mitochondria by mitophagy was observed in the KO cell line, after inducing mitophagy with CCCP and over-expression of Parkin (*Narendra et al., 2008*) (*Figure 5D*) together with an increase in S616 phosphorylation of Drp1, a phosphorylation associated with decreased fusion and increased fission (*Figure 5E–F*, *Figure 5—figure supplement 3*) (*Rambold et al., 2011*; *Kashatus et al., 2015*). Together our results demonstrate that VPS13A depleted cells show an apparent mitochondria phenotype consistent with decreased fusion, increased fission and impairment of mitophagy.

## VPS13A is associated with lipid droplets

In addition to a localization at areas were mitochondria and the ER are in close proximity, we observed that VPS13A is also appeared in a punctate and vesicular-shaped pattern. These vesicular-like structures did not represent mitochondria (*Video 1*). Using confocal microscopy with lipid droplets (LDs) specific dyes, BODIPY-FA or LipidTox red, we showed that the VPS13A positive structures co-localized with these dyes, indicating that these VPS13A positive vesicular-like structures were LDs (*Figure 6A*).

In order to elaborate further on this observation, cells were cultured under conditions that elicit LD biogenesis and oleic acid (OA), a fatty acid known to induce intracellular LD formation (*Wilfling et al., 2013*; *Thiel et al., 2013*; *Kassan et al., 2013*) was added to the cells. Cells expressing VPS13A-GFP were visualized at different times after OA induction. Before the addition of OA and under normal culturing conditions, a small amount of LDs were observed which were positive for VPS13A-GFP, in addition to the VPS13A-GFP signal present in the reticular pattern reflecting its distribution at the mitochondria-ER contact sites (*Figure 6B*, left panel). After 2 hr of exposure to OA, numerous LDs were formed and VPS13A-GFP was found at BODIPY-FA-positive LDs. Line scan analysis of individual large LDs at a high magnification revealed that VPS13A-GFP uniformly encircled them (*Figure 6C,C'*), indicating enrichment of VPS13A at the membrane and not at the interior of LDs, the ring-like VPS13A positive signal is most obvious at the periphery of larger LDs (such as after 120' OA, *Figure 6B*).

To corroborate these observations, we next investigated whether endogenous VPS13A was also enriched in fractions enriched with LDs. We thus analyzed the subcellular distribution of endogenous VPS13A by sucrose gradient fractionation of cells grown under normal conditions, starved for serum or exposed to OA for 24 hr (*Figure 6—figure supplement 1*). Western blot analysis of sucrose

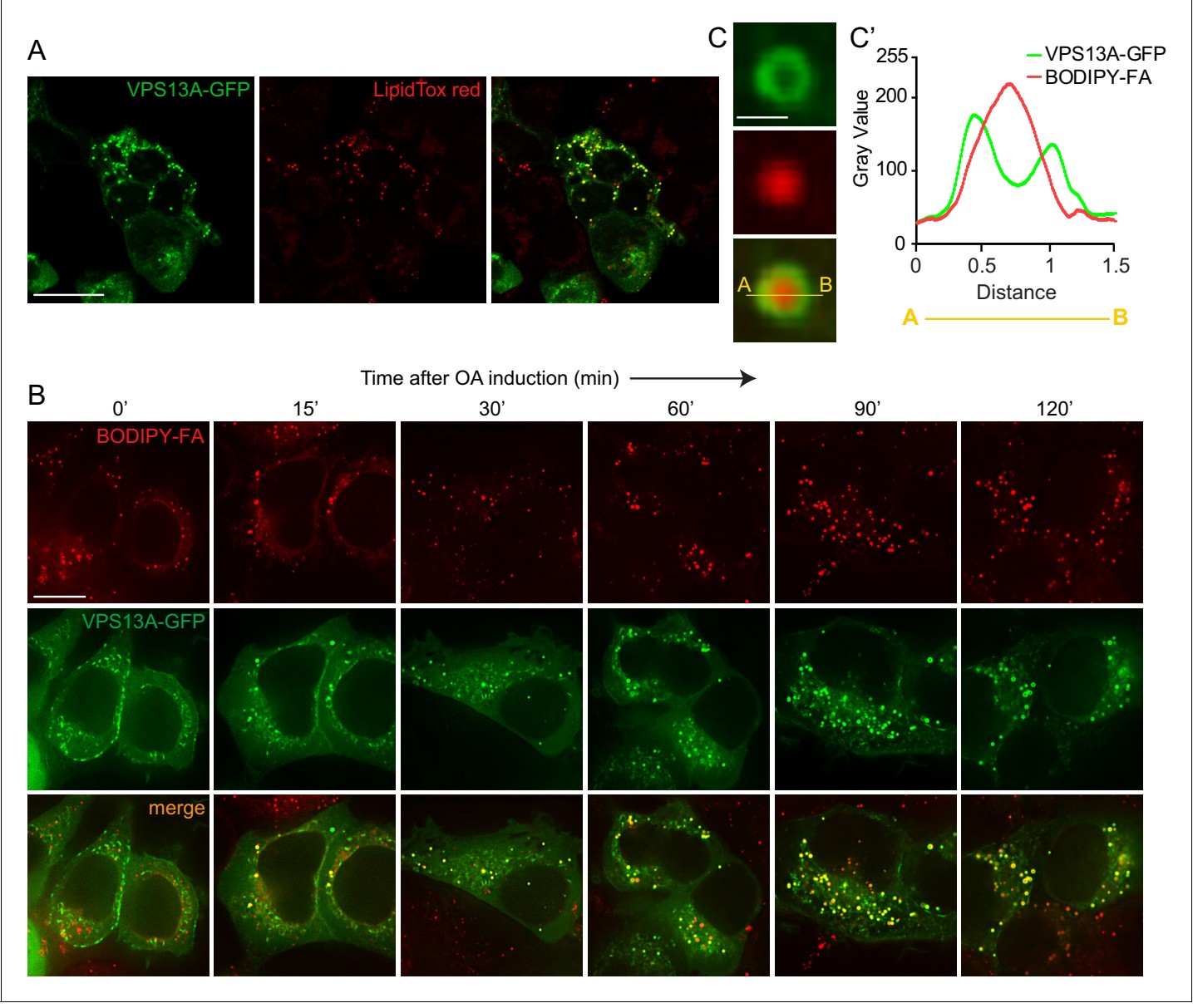

**Figure 6.** VPS13A decorates Lipid droplets. (**A**) HEK293T cells were transfected with VPS13A-GFP for 24 hr and Lipidtox red was used as a marker for LDs. (**B**) HEK293T cells transfected with VPS13A-GFP for 48 hr were pulsed with 1 µM BODIPY-FA (red) at 37°C for 30 min followed by a chase in medium containing 500 uM OA for 2 hr at 37°C. (**C**) A close-up image of a LD in a cell taken from B in vivo is shown. Line profile analysis across the LD showed the enrichment of the VPS13A-GFP signal on the periphery of the LD (**C'**). Scale bar = 1 µm. Scale bars = 10 µm (**A, B**) and 1 µm (**C**).

DOI: https://doi.org/10.7554/eLife.43561.023

The following figure supplement is available for figure 6:

**Figure supplement 1.** Endogenous VPS13A is enriched at fractions containing LDs upon OA induction Workflow of LDs isolation and sucrose gradient fractionation.

DOI: https://doi.org/10.7554/eLife.43561.024

gradient fractions revealed that VPS13A was mainly enriched in the heavier fractions under starvation (*Figure 7A,A' and A"*) and normal (*Figure 7—figure supplements 1–4*) growth conditions, and only a small portion (~4%) appeared in fraction 1, corresponding to LDs that floated on top of the sucrose gradient, which was identified using the Perilipin2 (PLIN2) as a specific LD marker protein. Part of PLIN2 was sequestered in the fractions with high density organelles that contained marker proteins such as VAP-A, EGFR and ATP-5A (*Figure 7A,A' and A"*, *Figure 7—figure supplement 1–*

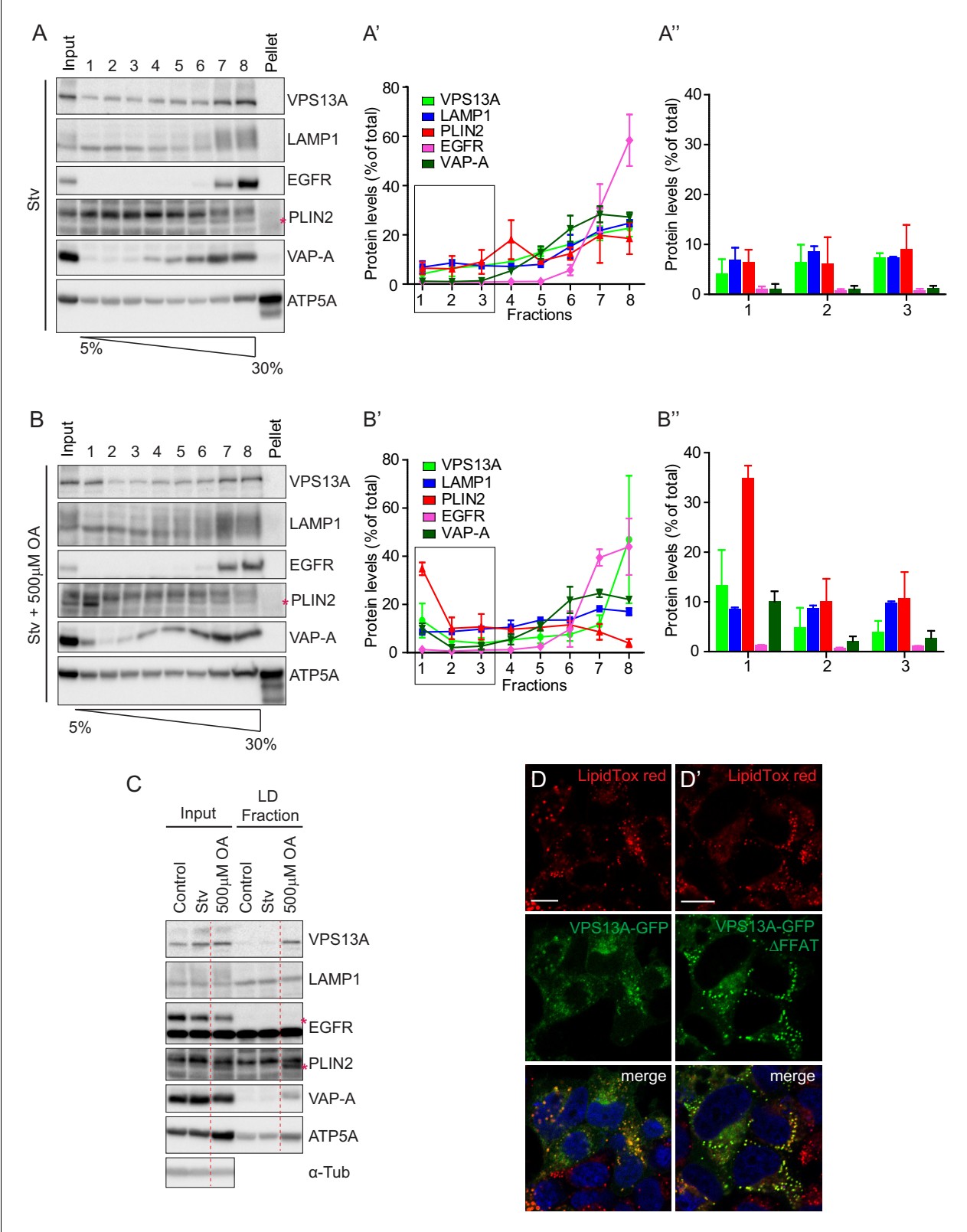

**Figure 7.** Endogenous VPS13A is enriched in LDs containing fractions. (**A**) FBS starved HeLa cells were processed as described in *Figure 6—figure supplement 1*. Fractions with equal amounts of proteins were processed for Western blot analysis and specific protein levels were detected using antibodies for VPS13A, LAMP1, EGFR, PLIN2, VAP-A and ATP5A. Quantification of protein band intensities in A was performed using ImageJ and plotted as percentage of the total (**A'**). A' shows a close-up of values of the top three light sucrose density fractions of A. In A'and A', error bars,
*Figure 7 continued on next page*

*Figure 7 continued*

mean ±s.e.m (n = 3). (B) FBS starved Hela cells were incubated with 500 μM OA and processed as described under A and as in *Figure 6—figure supplement 1*. Quantification of protein band intensities in B was performed using ImageJ and plotted as percentage of the total (B'). B' shows a close-up of values of the top three lowest sucrose density fractions. In B'and B', error bars, mean ±s.e.m (n = 3). (C) HeLa cells were either grown in complete medium (Control), FBS starved (Stv, as in A) or further incubated with 500 μM OA and processed as described in *Figure 6—figure supplement 1*. LDs were isolated from the top fraction. Equal amounts of proteins were resolved by Western Blot and detected using antibodies for VPS13A, LAMP1, EGFR, PLIN2, VAP-A, ATP5A and α-Tubulin. Specific bands are indicated with an asterisks D) Representative single stack image of HEK293T cells expressing VPS13A-GFP (D) or VPS13A-GFP $^{\Delta FFAT}$ (D'). Cells were incubated with 500 μM OA for 3 hr. LDs stained with LipidTox red. Scale bar = 10 μm (D).

DOI: https://doi.org/10.7554/eLife.43561.025

The following figure supplements are available for figure 7:

**Figure supplement 1.** Scan of original blots for *Figure 7*.

DOI: https://doi.org/10.7554/eLife.43561.026

**Figure supplement 2.** Endogenous VPS13A is enriched at fractions containing LDs upon OA induction.

DOI: https://doi.org/10.7554/eLife.43561.027

**Figure supplement 3.** Scan of original blots for *Figure 7—figure supplement 2*.

DOI: https://doi.org/10.7554/eLife.43561.028

**Figure supplement 4.** Endogenous VPS13A is enriched at fractions containing LDs upon OA induction.

DOI: https://doi.org/10.7554/eLife.43561.029

*4*), consistent with previous work showing that very minimal amount of LDs are formed under starvation conditions (*Kassan et al., 2013*). Induction of LD formation after incubation of cells with OA for 24 hr resulted in a shift in the distribution of endogenous VPS13A towards the LD fraction. As expected, PLIN2 was enriched in the top fraction consistent with the fact that LDs are formed in response to OA induction (*Figure 7B,B' and B"*, *Figure 7—figure supplements 1–4*). The distribution of the plasma membrane protein EGFR and the lysosomal protein LAMP1 was not affected upon OA induction or serum starvation (*Figure 7A–B"*, *Figure 7—figure supplements 1–4*). In addition, comparison of the amount of VPS13A in the LD fraction showed that VPS13A was partly concentrated in the LD fractions of OA fed cells. Addition of OA to starved cells increased the amount of VPS13A in the LD fraction (*Figure 7C*, *Figure 7—figure supplements 1–4*). Taken together, these data confirmed our observation that VPS13A is associated with LDs.

We then questioned whether the ER localization through VAP-A binding was important for the LDs localization of VPS13A. To do so, we expressed VPS13A-GFP$^{\Delta FFAT}$ in OA fed cells and showed that it was recruited to LDs similarly as WT VPS13A-GFP (*Figure 7D,D'*). This indicates that the FFAT motif of VPS13A is not required for its recruitment to LDs.

## VPS13A negatively affects lipid droplet size and motility

We investigated the role of VPS13A on LDs biology by studying the number of LDs in the presence and absence of VPS13A, and we compared the motility of VPS13A-positive and VPS13A-negative LDs. Under normal culturing conditions, VPS13A KO cells showed increased numbers of LDs (*Figure 8A–B*) compared to the parental control line. In addition, fluorescent activated cell sorting (FACS) quantification of the total Nile red intensity showed a significantly increased intensity in the absence of VPS13A (*Figure 8C*). VPS13A is not required for LD formation, because VPS13KO cells do contain LDs and OA induction in VPS13A KO cells resulted in an increase in LDs comparable to control cells (*Figure 8D*).

Live cell analysis was used to track individual LDs in VPS13A-GFP expressing cells. Visual examination showed that VPS13A-GFP positive LDs slowly and randomly oscillated. When these LDs were briefly dissociated from VPS13A-GFP, they directionally traveled faster and such motility was interrupted when VPS13A-GFP was again associated with the LD (*Video 3*). To further substantiate this, we recorded LDs in adjacent control (*Figure 8E,E'* cell 2) and VPS13A-GFP overexpressing (**Figure E, E'** cell 1) HEK293T cells, at two different times and quantified the LDs that did not move at this time interval. In VPS13A-GFP overexpressing cells, a larger fraction of the LDs showed an overlapping pattern compared to the non-transfected cells (*Figure 8E–F*), further suggesting that VPS13A overexpression reduces LD motility.

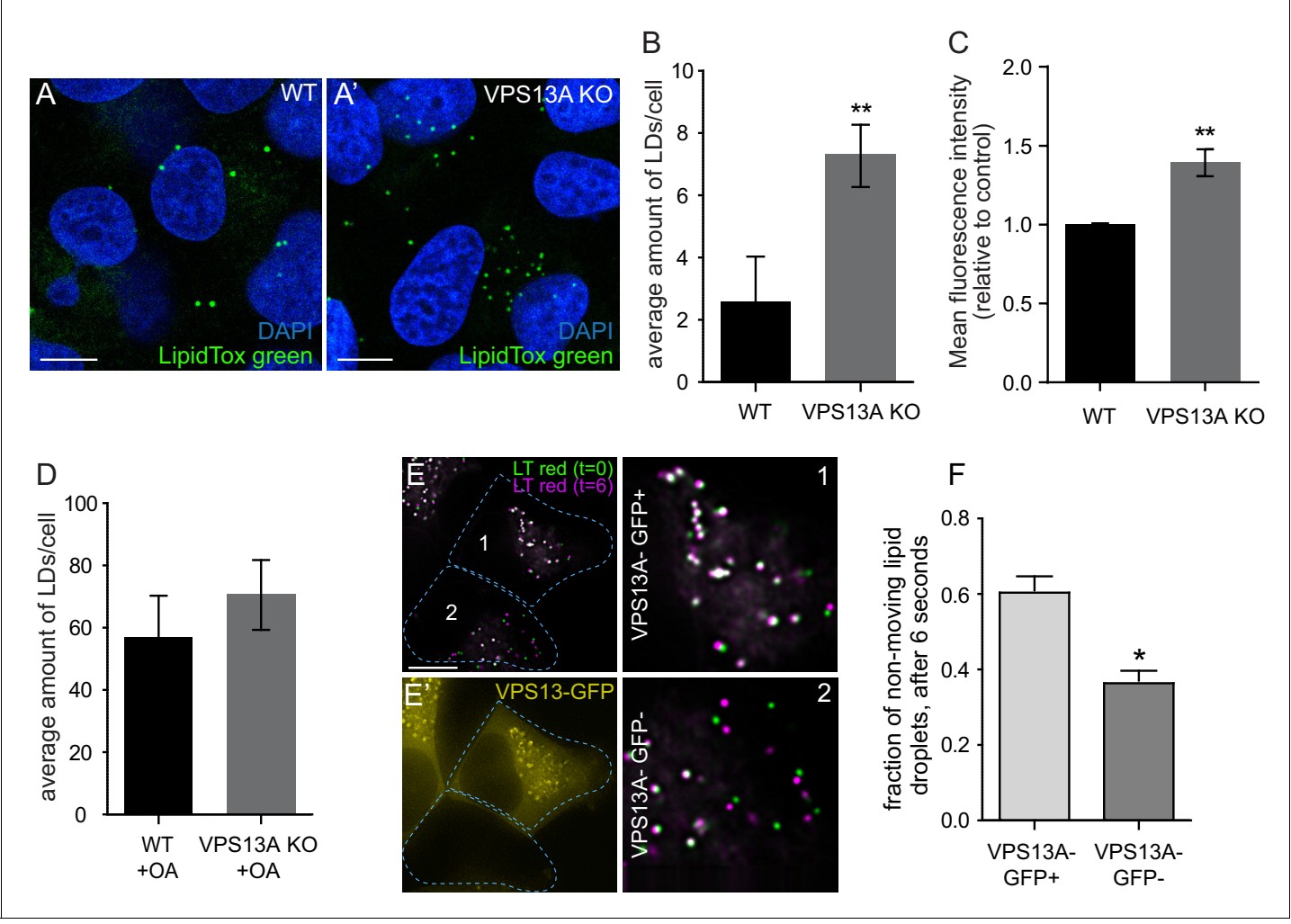

**Figure 8.** VPS13A negatively regulates LD mobility. (A) WT (A) and VPS13A KO MRC5 cells (A') were stained with LipidTox green for LDs (green) and the nuclear marker DAPI (blue) and imaged by confocal microscopy. (B) Quantification of LD numbers in A. Error bars, mean ±s.e.m (n = 3), two-tailed unpaired Student's t-test was used (*p≤0.05, **p≤0.01). (C) WT and VPS13A KO MRC5 cells were stained with Nile red and intensity was measured using FACS. Error bars, mean ±s.e.m (n = 3), two-tailed unpaired Student's t-test was used (*p≤0.05, **p≤0.01). (D) WT and VPS13A KO MRC5 cells were exposed to 500 µM OA for 16 hr. Afterwards cells were stained with LipidTox green to visualize LDs and LD numbers were quantified. Error bars, mean ±s.e.m (n = 3), two-tailed unpaired Student's t-test was used. (E) HEK293T cells were transfected with VPS13A-GFP and stained with LipidTox red to visualize LDs in vivo. Images with a time interval of 6 s were recorded of VPS13-GFP positive (cell 1) and adjacent VPS13-GFP negative (cell 2) cells. The locations of LDs at t = 0 are indicated in green, the locations of the same LDs at t = 6 s are indicated in magenta (E). If the LD did not move between time frames, the overlapping signal (green and magenta) is white. The VPS13A signal is shown in E': Cell one is transfected with VPS13A-GFP; Cell two is a non-transfected cell. (F) Quantification of the fraction of non-moving (white) LDs compared to the total number of LDs in VPS13A-GFP positive or VPS13A-GFP negative cells. Error bars, mean ±s.e.m, two tailed unpaired Student's t-test was used (*p≤0.05). Scale bars = 10 µm (A, A', E,).
DOI: https://doi.org/10.7554/eLife.43561.030

In summary, the presence of VPS13A on LDs negatively influenced their motility and when LDs temporarily did not contain VPS13A, they showed faster directional motility. In the absence of VPS13A increased LD numbers are present, strongly indicating a role of VPS13A in LD related processes.

## Eyes of *Drosophila Vps13* mutants show an increase in LDs

Previously it has been demonstrated that in pigment cells (glia cells) of *Drosophila* eyes LDs can be formed in response to various stressors occurring in neuronal cells (*Liu et al., 2015*). In order to investigate the role of VPS13A in LD related processes in a multicellular organism, we investigated

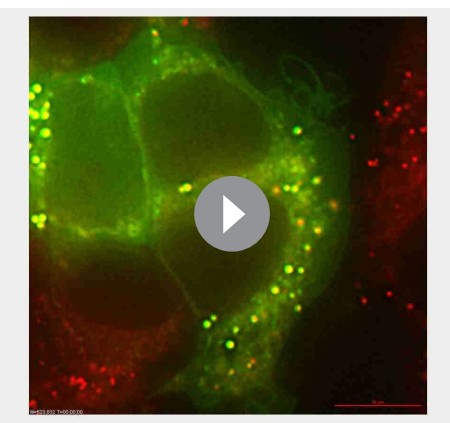

**Video 3.** HEK 293 T cells overexpressing VPS13-GFP were incubated with 500 uM for 3 hr. LDs were stained with LipidTox red to visualize and time lapse images were taken every 600 milliseconds. The video is played at 10 frames per second.

DOI: https://doi.org/10.7554/eLife.43561.031

LDs in eyes of the available and established *Drosophila Vps13* mutant (*Vonk et al., 2017*). *Drosophila* Vps13 is most similar to human VPS13A and VPS13C (*Velayos-Baeza et al., 2004*). Homozygous mutants show a decreased life span, impaired locomotor function upon ageing, impaired protein homeostasis and large brain vacuoles (*Vonk et al., 2017*). Examination of the eyes using Nile red to visualize LDs showed that 5 day old *Vps13* mutants have increased numbers of LDs compared to wildtype (*Figure 9A–C'*). Overexpression of human *VPS13A* in the mutant background (*Figure 9D–E*, *Figure 9—figure supplement 1*) rescued the phenotype back to normal. These data indicate that *Drosophila* Vps13 and human VPS13A share functional properties.

## Discussion

Our biochemical and localization studies show that human VPS13A is a peripheral membrane protein present, at least, at two distinct subcellular localizations: at sites where mitochondria and the ER are in close proximity and VPS13A is localized at the surface of LDs. These results confirm early observations obtained from overexpression of human VPS13A in mammalian cells and identify the characteristic 'vesicular-like' structures as LDs (*Velayos-Baeza et al., 2008*). The peripheral membrane characteristics of VPS13A are shared by the other human VPS13 proteins (B, C and D), the yeast and the *Drosophila* Vps13 protein (*Lesage et al., 2016*; *Brickner and Fuller, 1997*; *Vonk et al., 2017*; *Velayos-Baeza et al., 2008*; *Seifert et al., 2011*), suggesting a common feature of VPS13 proteins.

### VPS13A is localized at ER-mitochondria contact sites

The association of VPS13A with the ER is established via its FFAT domain which binds to the ER residing protein VAP-A. VAP-A/B proteins have been extensively characterized as a hub when the ER establishes membrane contacts with other organelles including endosomes, mitochondria, peroxisomes, plasma membrane and Golgi (*Alpy et al., 2013*; *Eden et al., 2016*; *Costello et al., 2017*; *Hua et al., 2017*; *Stoica et al., 2014*; *Gomez-Suaga et al., 2017*; *Mesmin et al., 2013*; *Stefan et al., 2011*; *Rocha et al., 2009*; *Dong et al., 2016*). Our results showed that VPS13A also interacts with VAP-B in a FFAT dependent manner (*Figure 4—figure supplements 2–3*), consistent with the fact that VAP-A and VAP-B functions are often redundant (*Dong et al., 2016*).

The association of VPS13A with mitochondria is mediated via the C-terminal domain. In addition, fractionation studies show that VPS13A co-fractionates with TOMM20, a protein localized at the outer membrane of mitochondria. Our observed interaction between VPS13A and VAP-A in a FFAT-dependent manner and our reported localization at the ER-mitochondria contact sites is consistent with localization studies recently reported by Kumar *et al* (*Kumar et al., 2018*).

### VPS13A depleted cells show mitochondria abnormalities

We further show that ER-mitochondria contact sites are decreased in VPS13A depleted cells, consistent with results by Kumar et al, which demonstrate that upon overexpression of VPS13A an increase in ER-mitochondria contact sites is observed. Our data and the data by Kumar et al are in line with studies in yeast demonstrating that Vps13 is present at various organelle contact sites and is required for ER-mitochondria contact sites, all pointing to a conserved function of VPS13A at these sites. Our reported mitochondria phenotypes (less elongated and a decreased mitophagy capacity) in the VPS13A depleted cells could all be explained by abnormal lipid composition of mitochondria membranes. A possible defect in lipid transfer between ER and mitochondria due to VPS13A depletion is in line with results from Kumar et al., who demonstrated in vitro that the N-terminal part of

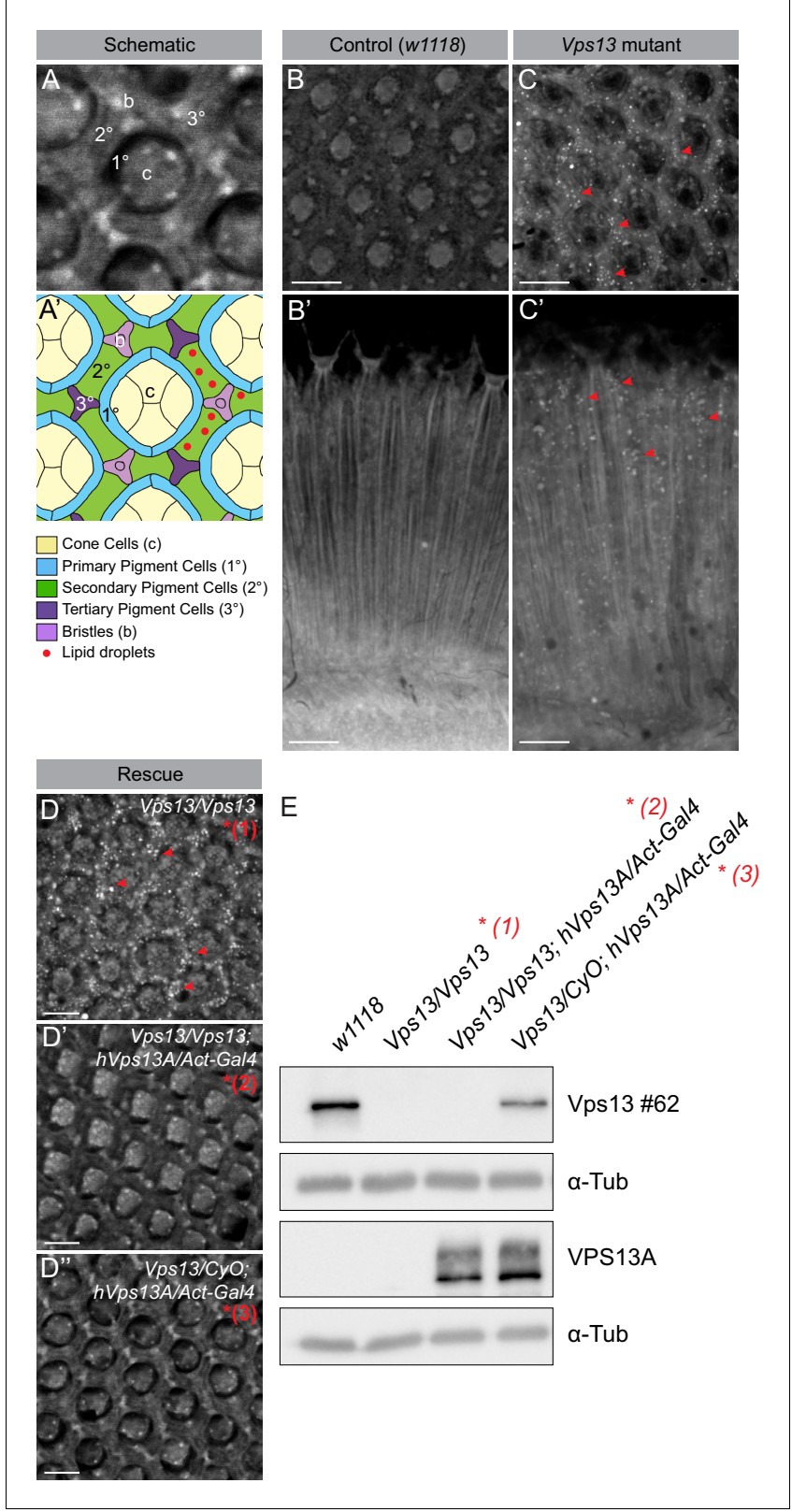

**Figure 9.** Vps13A mutants show a lipid droplet phenotype in the Drosophila adult eye, which can be rescued by ectopic expression of the human VPS13A. (**A**) Optical section (**A**) and schematic (**A'**) of a *Drosophila* adult ommatidium, taken at the height of the cone cells (*Ready, 1989*). Cone cells (**c**), pigment cells (1°, 2°, 3°) and bristles (**b**) are indicated in different colors. Naturally occurring or (in mutants) ectopically accumulating LDs (red

*Figure 9 continued on next page*

*Figure 9 continued*

circles) are found in the pigment cells (*Liu et al., 2015*; *Liu et al., 2017*). (**B,C**) Optical cross-section and longitudinal section through the adult eye of *Drosophila* control (B/B') and *Vps13* homozygous mutant flies (C/C') at day 5 past eclosion. Nile Red was used to reveal the presence of LDs (red arrow heads) in the pigment cells. (**D**) Optical cross-sections through the adult eye of *Vps13* homozygous mutant flies, control flies and *Vps13* homozygous mutant flies expressing human VPS13A at day 3 after eclosion. Nile Red was used to detect LDs. D: *Vps13/Vps13,* (=*Vps13* homozygous mutant). D': *Vps13/Vps13;hVPS13A/Act-Gal4* (=*Vps13* homozygous mutant expressing human VPS13A). *D': Vps13/+;UAS-hVPS13A/Act-Gal4* (heterozygous for *Vps13* expressing human VPS13A). (**E**) Western blot to demonstrate the absence of *Drosophila* Vps13 in mutant flies and the expression of human VPS13A in the rescued *Drosophila Vps13* mutant background. Samples marked with a red asterisk were used for the Nile Red staining in the rescue experiment (**D–D''**). Scale bars = 10 µm (**B–D**).
DOI: https://doi.org/10.7554/eLife.43561.032
The following figure supplement is available for figure 9:

**Figure supplement 1.** Scan of original blots for *Figure 9*.
DOI: https://doi.org/10.7554/eLife.43561.033

yeast Vps13, which is highly similar to human VPS13A, is able to transfer lipids between two membranes. Together, these data favor a model in which human VPS13A plays a role in tethering ER to the outer membrane of mitochondria to create areas of close proximity and to enable transfer of lipids between these membranes via the VPS13A N-terminal domain (*Figure 10A,C*).

## VPS13A is localized at the surface of lipid droplets

In addition to the localization at ER-mitochondria contact sites, VPS13A localizes to the periphery of LDs in a FFAT-independent manner, consistent with the recent report from Kumar et al (*Kumar et al., 2018*). Under circumstances of increased fatty acid uptake, more LDs accumulate in cells and thereby more VPS13A positive LDs are observed. The origin of LD-associated VPS13A could be either newly synthesized VPS13A or protein relocated from the already available VPS13A pool, mainly at the ER-mitochodria contact sites, more in depth studies are required to address this point. Bean et al. (*Bean et al., 2018*) have recently shown in yeast that different adaptor proteins present at specific subcellular locations compete for binding to Vps13. Organelle-specific VPS13A adaptor proteins may be present as well in mammalian cells; LD specific adaptor proteins would increase in conditions when LDs are increased, resulting in enhanced competition for VPS13A which could possibly be relocated to LDs from other sub-cellular locations. This explanation (*Figure 10B, D*) is in line with our observation of increased levels of VPS13A in fractions containing LDs in cells with an increased amount of LDs. Different VPS13 members may have their own specific adaptor proteins which would explain their different reported localizations, such as VPS13B at the Golgi (*Seifert et al., 2011*) or VPS13C at endosomes (*Kumar et al., 2018*). Conversely, since different VPS13 proteins can localize at the same organelles, such as VPS13A and VPS13C in LDs and mitochondria (*Lesage et al., 2016*; *Kumar et al., 2018*; *Yang et al., 2016*; this report), it is also possible that the same adaptor protein could bind several VPS13 proteins.

## VPS13A influences lipid droplet motility

LDs have long been considered as inert lipid inclusions and studies of their biology were constrained (*Gluchowski et al., 2017*). Evidence is now accumulating that LDs are far from being only fat depots as they are decorated by a large number of proteins that regulate their formation, destruction and communication with other organelles (*Kassan et al., 2013*; *Thiam and Forêt, 2016*; *Salo et al., 2016*; *Wang et al., 2016*; *Bi et al., 2014*; *Krahmer et al., 2011*; *Kory et al., 2015*; *Cermelli et al., 2006*). Given the described functions of VPS13A in tethering ER-mitochondria membranes and transferring lipids, it could be expected that VPS13A at LDs is probably performing a comparable function. Kumar et al demonstrated that LDs decorated with VPS13A are surrounded by ER and, therefore, most likely VPS13A could be at contact sites between LDs and ER (*Figure 10B,D*). VPS13A influences the motility of LDs, a feature reminiscent of identified proteins regulating dynamics of endosomal vesicles. Endosomal movement is halted when endosomes make contacts with the ER (*Raiborg et al., 2015*) and movement of peroxisomes is increased upon loss of the VAP-ACBD5 tethering complex (*Costello et al., 2017*; *Hua et al., 2017*). Consistent with this, we show that

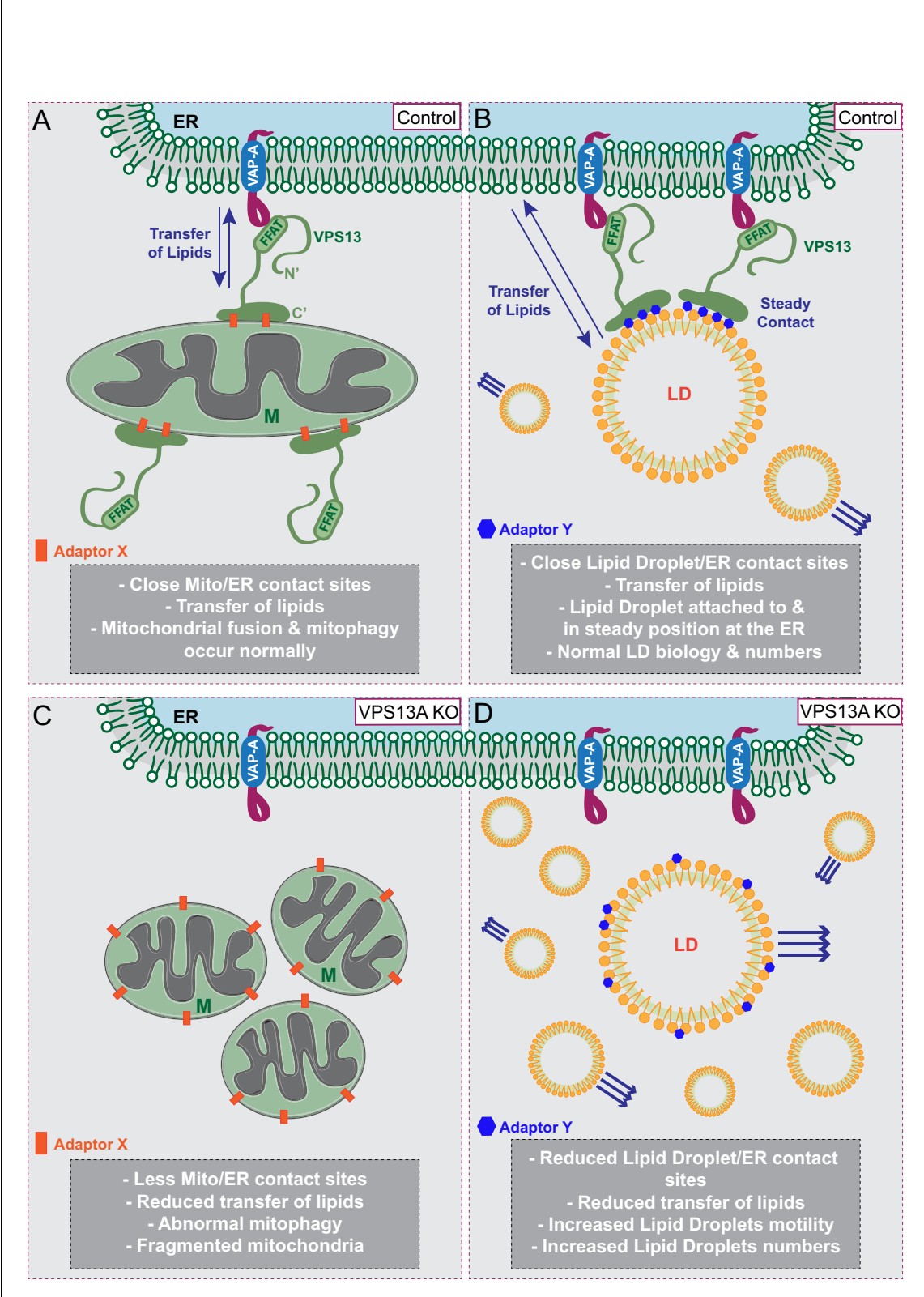

**Figure 10.** Proposed model for VPS13A function. (**A**) Under normal growth conditions VPS13A is localized at the ER-mitochondria contact sites where it is anchored to VAP-A through its FFAT domain and via its C-terminal region it is associated with mitochondria, most likely via mitochondria specific adaptor proteins. VPS13A at this location may facilitate the transfer of lipids between ER and mitochondria and mitochondria fusion and mitophagy occur normally. (**B**) Under normal conditions VPS13A is also associated to LD, an association mediated via LD specific adaptor proteins. Via VPS13A LD

*Figure 10 continued on next page*

*Figure 10 continued*

are associated to the ER and VPS13A facilitate the transfer of lipids between ER and LDs. The VPS13A mediated ER-lipid connection halts LD movement. (C) Depletion of VPS13A leads to impaired lipid transfer between ER and mitochondria, leading to abnormal function of mitochondria which become less elongated. (D) Depletion of VPS13A also leads to disconnection of LD and the ER, leading to increased movement and reduced degradation of LD, resulting in increased LD numbers.

DOI: https://doi.org/10.7554/eLife.43561.034

VPS13A negatively influences LD motility and LDs are more fixed under conditions of VPS13A overexpression.

## VPS13A depleted cells and *Drosophila Vps13* mutants show increased amount of lipid droplets

Increased numbers of LDs in VPS13A depleted cells can be explained because in the absence of VPS13A the association with the ER may be reduced and lipid transfer decreased. This in turn could lead to disruption of LD turnover processes such as lipophagy and release of LD content to other organelles (*Rambold et al., 2015*; *Kaushik and Cuervo, 2015*). Homozygous *Drosophila Vps13* mutants also show an increase in LDs, which could be explained by a combination of impaired mitochondria function and abnormal LD turnover capacity. It has been reported that, in response to impaired mitochondria function in neuronal cells of the *Drosophila* eye, ROS levels increase and lipids are transferred from neurons to glia cells where LDs transiently form (*Liu et al., 2015*; *Liu et al., 2017*). An increase in LDs in glia cells in response to impaired mitochondria functioning is also observed in neurodegenerative mouse models (*Liu et al., 2015*; *Liu et al., 2017*). Thus, it is possible that the increased numbers of LDs in glia cells of the fly *Vps13* mutant eyes could be caused by an initial impairment in mitochondrial function.

## VPS13A and ChAc

The question remains why loss of VPS13A leads to ChAc, a movement disorder mostly presenting in the third decade of the patient's life. Impairment of mitochondria processes such as fusion and mitophagy could explain the neurodegeneration observed in ChAc patients, since impairment of these processes has been largely linked to neurodegeneration (*Ryan et al., 2015*). In addition, impairment of LD related processes could explain neurodegeneration as well since LD abnormalities are associated with several neurodegenerative diseases such as hereditary spastic paraplegias (*Inloes et al., 2014*), Huntington's disease (*Martinez-Vicente et al., 2010*), and Parkinson's disease (*Outeiro and Lindquist, 2003*). The role of LD in the adult central nervous system is largely unknown. It may be possible that in ageing ChAc patients oxidative stress builds up due to impaired mitochondria functions and LDs form and accumulate because of a compromised turnover due to decreased contact sites with their target organelles. Gradually increasing numbers of large LDs in an aging organism may form physical obstructions that could eventually hamper cellular functions of glia and their neighboring neuronal cells. It is also well possible that overall lipid homeostasis and other metabolic pathways are imbalanced in ChAc, leading to neurodegeneration in an ageing organism. Since LDs have not been studied in ChAc models or in material derived from ChAc patients, these possible 'disease mechanisms' are only hypotheses which would require further experimental data to be properly tested, leaving this field largely open for future research.

## Materials and methods

**Key resources table**

| Reagent type (species) or resource | Designation | Source or reference | Identifiers | Additional information |
|---|---|---|---|---|
| Antibody | Flag (rabbit polyclonal) | Sigma | F7425 | IF (1:500) |
| Antibody | Myc (mouse monoclonal) | Enzo Life Science | ADI-MSA-110-F | IF (1:500) WB (1:1000) |

*Continued on next page*

*Continued*

| Reagent type (species) or resource | Designation | Source or reference | Identifiers | Additional information |
|---|---|---|---|---|
| Antibody | TOMM20 (mous monoclonal) | BD biosciences | 612278 | IF (1:200) WB (1:1000) |
| Antibody | Normal Goat IgG (goat polyclonal) | Santacruz | sc-2028 | IP (1:200) |
| Antibody | Normal rabbit IgG (rabbit polyclonal) | Santacruz | sc-2027 | IP (1:200) |
| Antibody | VAP-A (goat polyclonal) | Santacruz | sc-48698 | IP (1:100) WB (1:1000) |
| Antibody | VAP-B (rabbit polyclonal) | Sigma | HPA013144 | IP (1:100) WB (1:1000) |
| Antibody | VPS13A (rabbit polyclonal) | Sigma | HPA021652 | IP (1:100) WB (1:1000) |
| Antibody | ATP5A (mouse monoclonal) | Abcam | ab14748 | WB (1:5000) |
| Antibody | a-Tubulin (mouse monoclonal) | Sigma | T5168 | WB (1:5000) |
| Antibody | EGFR (rabbit polyclonal) | Santacruz | SC-03-G | WB (1:1000) |
| Antibody | GAPDH (mouse monoclonal) | Fitzgerald | 10R-G109A | WB (1:10000) |
| Antibody | GFP (mouse monoclonal) | Clontech | 632381 | WB (1:5000) |
| Antibody | GST (mouse monoclonal) | Santacruz | sc-138 | WB (1:1000) |
| Antibody | LAMP1 (mouse monoclonal) | Abcam | ab25630 | WB (1:1000) |
| Antibody | DRP1 (rabbit monoclonal) | cell signaling | 8570 s | WB (1:500) D6C7 |
| Antibody | pDRP1 (rabbit polyclonal) | cell signaling | 3455 s | WB (1:1000) ser616 |
| Antibody | PLIN2 (rabbit polyclonal) | Abcam | ab78920 | WB (1:1000) |
| Antibody | RAB7 (mouse monoclonal) | Abcam | ab50533 | WB (1:1000) |
| Antibody | Vps13 #62 (rabbit polyclonal) | PMID:28107480 | | WB (1:1000) |
| Antibody | VPS13A (rabbit polyclonal) | Sigma | HPA021662 | WB (1:1000) |
| Antibody | VPS13A (H-102) (rabbit polyclonal) | Santacruz | sc-367262 | WB (1:1000) |
| Antibody | VPS13C (rabbit polyclonal) | Sigma | HPA043507 | WB (1:1000) |
| Other | Nile Red | Thermo Fisher Scientific | N1142 | FACS (1:500) |
| Other | BODIPY-FA | Thermo Fisher Scientific | D3835 | IF 1 μM |
| Other | LipidTox-green | Thermo Fisher Scientific | H34475 | IF (1:200) |
| Other | LipidTox-red | Thermo Fisher Scientific | H34476 | IF (1:200) |

*Continued on next page*

*Continued*

| Reagent type (species) or resource | Designation | Source or reference | Identifiers | Additional information |
|---|---|---|---|---|
| Other | Mitotracker Orange | Thermo Fisher Scientific | M-7510 | 100 nM (live) and 200 nM (fixed) |
| Other | Mitotracker Red | Thermo Fisher Scientific | M-7512 | 100 nM (live) and 200 nM (fixed) |
| Other | Nile Red | Thermo Fisher Scientific | N1142 | IF (1:1000) |
| Other | DAPI | Thermo Fisher Scientific | 62247 | 0.2 µg/ml |
| Recombinant DNA reagent | Lamp1-GFP | Addgene | 34831 | |
| Recombinant DNA reagent | mCherry-FYCO1 | PMID:25855459 | | |
| Recombinant DNA reagent | GFP-Rab5 Q79L | Addgene | 28046 | |
| Recombinant DNA reagent | GFP-Rab7 Q67L | Addgene | 28049 | |
| Recombinant DNA reagent | BFP-Sec61B | Addgene | 49154 | |
| Recombinant DNA reagent | mCherry-Sec61B | Addgene | 49155 | |
| Recombinant DNA reagent | peGFP-C1 | Clontech | discontinued | |
| Recombinant DNA reagent | peGFP-N1 | Clontech | 6085–1 | |
| Recombinant DNA reagent | VPS13-GFP (FL) | this paper | | Progentiors:PCR VPS13-myc and pEGFP-N1; VPS13-myc |
| Recombinant DNA reagent | VPS13-Myc (FL) | PMID:28107480 | | |
| Recombinant DNA reagent | VPS13-GFP- DFFAT | this paper | | mutagenesis on VPS13-GFP |
| Recombinant DNA reagent | VPS13-GFP 2–854 | this paper | | Progentiors: PCR VPS13-GFP; pEGFP-C1 |
| Recombinant DNA reagent | VPS13-GFP 835–1700 | this paper | | Progentiors: PCR VPS13-GFP; pEGFP-C1 |
| Recombinant DNA reagent | VPS13-GFP 855–1700 | this paper | | Progentiors: PCR VPS13-GFP; pEGFP-C1 |
| Recombinant DNA reagent | VPS13-GFP 2003–2606 | this paper | | Progentiors: PCR VPS13-GFP; pEGFP-C1 |
| Recombinant DNA reagent | VPS13-GFP 2615–3174 | this paper | | Progentiors: PCR VPS13-GFP; pEGFP-C1 |
| Recombinant DNA reagent | pGEX5×2 | GE Healthcare | 28954554 | |
| Recombinant DNA reagent | GST-FFAT | this paper | | Progentiors: oligo FFAT domain; pGEX5×2 |
| Recombinant DNA reagent | GST-VPS13A (2-834) | this paper | | Progentiors: PCR VPS13-GFP; pGEX5×2 |

*Continued on next page*

*Continued*

| Reagent type (species) or resource | Designation | Source or reference | Identifiers | Additional information |
|---|---|---|---|---|
| Recombinant DNA reagent | GST-VPS13A (2-854) | this paper | | Progentiors: PCR VPS13-GFP; pGEX5×2 |
| Recombinant DNA reagent | GST-VPS13A (2–854/D845A) | this paper | | mutagensis on GST-VPS13 (2-854) |
| Recombinant DNA reagent | GST-VPS13A (835–1700) | this paper | | Progentiors: PCR, VPS13-GFP; pGEX5×2 |
| Recombinant DNA reagent | pET28a | EMD Biosciences | 69864–3 | |
| Recombinant DNA reagent | GFP-VAP-A | this paper | | Progentiors: PCR pET28a-VAP-A; pEGFP-C1 |
| Recombinant DNA reagent | mCherry-VAP-A | this paper | | Progentiors: PCR pET28a-VAP-A; mCherry-tubuline |
| Recombinant DNA reagent | pET28a-VAPA | this paper | | Progentiors: PCR cDNA Hek293T; pET28a |
| Recombinant DNA reagent | SPLICS$_s$ | PMID: 29229997 | | |
| Recombinant DNA reagent | SPLICS$_L$ | PMID: 29229997 | | |
| Recombinant DNA reagent | OMM-GFP$_{1-10}$ | PMID: 29229997 | | |
| Recombinant DNA reagent | FLAG-Parkin | PMID: 12937272 | | |
| Recombinant DNA reagent | pSpCas9(BB)—2A-Puro (PX459) | Addgene | 48139 | |
| Recombinant DNA reagent | mCherry-tubuline | PMID: 15558047 | | |
| Cells (human) | Hek293T | ATCC | CRL-3216 | |
| Cells (human) | HeLa S3 | ATCC | CCL-2.2 | |
| Cells (human) | U2OS | ATCC | HTB-96 | |
| Cells (human) | MRC5 WT | (MRC-5 SV2) ECACC 84100401 PMID: 6313714 | | |
| Cells (human) | MRC5 Clone 4 MRC5-SV2_A01-01t_A2b | A. Velayos-Baeza | | |
| Chemical compound, drug | Oleic acid | Sigma | O3008 | |
| Chemical compound, drug | Thapsigargin | Merck Millipore | 586005 | |
| Chemical compound, drug | Carbonyl cyanide 3-chlorophenyl hydrazone (CCCP) | Sigma | C2759 | |
| Chemical compound, drug | Proteinase K (recombinant), PCR grade | Fermentas | EO0491 | |
| Commercial assay or kit | Gibson assembly master mix | NEB | E2611 | |
| Chemical compound, drug | HBSS, calcium magnesium | Thermo Fisher Scientific | 14025092 | |

*Continued on next page*

*Continued*

| Reagent type (species) or resource | Designation | Source or reference | Identifiers | Additional information |
|---|---|---|---|---|
| Commercial assay or kit | QuickChange Site Directed Mutagenesis Kit | Agilent | 200519 | |
| Commercial assay or kit | Glutathione Sepharose 4B (10 ml) | GE healthcare | 17-0756-01 | |
| Commercial assay or kit | GFP-Trap_MA | Chromotek | gtma-20 | |
| chemical compound, drug | polyethy lenimine (PEI) | Polysciences | 23966 | |
| Commercial assay or kit | protein A/G plus agarose beads | Santa Cruz | sc-2003 | |
| Genetic reagent (*D. melanogaster*) | w1118 | Bloomington Drosophila Stock Center | 3605 | FlyBase symbol: w[1118] |
| Genetic reagent (*D. melanogaster*) | VPS13 (c03628) | Harvard | c03628 | FlyBase symbol: PBac{PB}Vps 13c03628/CyO |
| Genetic reagent (*D. melanogaster*) | hVPS13 | PMID:28107480 | | |
| Genetic reagent (*D. melanogaster*) | Act-Gal4 | Bloomington Drosophila Stock Center | 3954 | y1w*;P{w[+mC]=Act5 C-GAL4}17bFO1/TM6B, Tb |

## Cell culture and transfection

HeLa, U2OS and HEK293T cells (all cell lines were obtained from ATCC, see Key Resources table) and are mycoplasma negative (GATC Biotech GA, Konstanz, Germany). MRC-5 SV2 cells (SV40-immortalized human male fetal lung fibroblasts), here referred to as MRC5, were initially obtained from ECACC (# 84100401), and were tested negative for mycoplasma. Cells were cultured in Dulbecco's modified eagle medium (DMEM, Gibco or Sigma) containing 10% Fetal Bovine Serum (FBS, Greiner Bio-one) and Penicillin/Streptomycin (Gibco) in 5% $CO_2$ at 37°C. Plasmid transfections of HeLa and U2OS cells were done using polyethylenimine 1 µg/ml(PEI, Polysciences) in 1:1 concentration. For procedures that required overexpression of full length VPS13A-GFP or VPS13A-Myc, HEK293T cells were transfected using the Calcium Phosphate precipitation method. In both cases cells were analyzed 24 or 48 hr after transfection and medium was refreshed 24 hr after transfection.

Oleic acid (OA, Sigma) was added at indicated concentrations for different time points. Thapsigargin (Merck Millipore) was added in indicated concentrations for 6 hr. Prior to HBSS (Thermo Fischer scientific) treatment, cells were washed 1x with HBSS and then incubated for 5 hr (pDRP1 determination) or for 16 hr (to assay mitochondria morphology) at 37°C in 5% $CO_2$.

## Plasmids and constructs

The full-length cDNA of the human *VPS13A* gene, variant 1A, was obtained as previously described (*Velayos-Baeza et al., 2004*; *Vonk et al., 2017*) and sub-cloned into pcDNA4-TO-mycHis (Invitrogen) to generate pcD13A-1A-mH, for expression in mammalian cells of VPS13A with a C-terminal myc +His tag (here referred to as VPS13A-Myc). A *XhoI-PciI* fragment of this plasmid, containing the myc-His tags and the zeocin selection marker, was replaced by a *XhoI-Eco*O109I fragment from pEGFP-N1 vector (Clontech), including EGFP and the kanamycin/neomycin selection marker, to generate pcD13A-1A-EGFP for expression of VPS13A with a C-terminal EGFP tag (here referred to as VPS13A-GFP). To generate the GFP-VPS13A constructs 2–854, 835–1700 and 855–1700, the correspond fragments were amplified by PCR from the full length VPS13A plasmid and inserted in to pEGFP-C1 (Clontech) via BamHI and XhoI restriction sites. To generate the GFP-VPS13A constructs 2003–2606 and 2615–3174, the corresponding fragments were amplified by PCR from the full length VPS13A plasmid and inserted in to the BamHI/KpnI site pEGFP-C1 with the Gibson assembly kit (NEB) according to the manufactures instructions. To generate the GST-VPS13A constructs 2–835, 2–854 and 835–1700, the fragments were amplified by PCR from the full length VPS13A plasmid

and inserted into pGEX5×2 (GE Healthcare) via SalI and NotI restriction sites. To generate the GST-VPS13A (2–854/D845A) and VPS13A-GFP$^{\Delta FFAT}$, a mutagenesis was performed on GST-VPS13A (2-854) or VPS13-GFP respectively, with the QuickChange Site Directed mutagenesis kit (Agilent) according to the manufacturer's protocol. To obtain GST-FFAT, oligonucleotides encoding the FFAT domain in human VPS13A (AA 842–848), flanked with SalI and 3' NotI sites, were synthesized, annealed and inserted into pGex5×2 via SalI and NotI restriction sites. To generate His-VAP-A, human VAP-A was amplified by PCR from HEK293 cDNA and inserted into pET28a (EMD Biosciences) via NdeI and BamHI restriction sites. To obtain GFP-VAP-A, VAP-A was amplified by PCR from the His-VAP-A plasmid and inserted into pEGFP-C1 (Clontech) via EcoRI and BamHI restriction sites. To obtain mCherry-VAP-A, tubulin in pcDNA3.1-mCherry-Tubulin (kind gift from B. Giepmans, (Shaner et al., 2004) was replaced by a VAP-A PCR fragment via BspEI and XhoI restriction sites. All restriction enzymes used where purchased from New England BioLabs (NEB). BFP-Sec61B (Addgene plasmid #49154) and mCherry-Sec61B (Addgene plasmid #49155) were kind gifts from Gia Voeltz (Zurek et al., 2011). GFP-Rab5 Q79L (Addgene plasmid #28046) and GFP-Rab7 Q67L (Addgene plasmid #28049) were kind gifts from Qing Zhong (Sun et al., 2010). LAMP1-mGFP (Addgene plasmid # 34831) was a kind gift from Esteban Dell'Angelica (Falcón-Pérez et al., 2005). (Raiborg et al., 2015). Plasmid pSpCas9(BB)−2A-Puro (PX459) (Addgene plasmid #48139) was a gift from Feng Zhang (Ran et al., 2013). mCherry FYCO1 was a kind gift from Harald Stenmark (Raiborg et al., 2015). OMM-GFP$_{1-10}$, SPLICS$_s$ and SPLICS$_L$ were a kind gift from Tito Cali (Cieri et al., 2018) FLAG-Parkin was a kind gift from Helen Ardley (Ardley et al., 2003).

| Primers | | Sequence 5'−3' |
|---|---|---|
| GFP-VPS13A (2-854) | FW | AATTGCTCGAGAAGGCGGCGTTTTCGAGTCGGTGGTCGTGGAC |
| | Rev | GGCCAAGGATCCAGGTTCTTCCAAGGGACTACAT |
| GFP-VPS13A (835–1700) | FW | GATCTCTCGAGGGGGCGGCTCTGAAGATGATTCAGAGGAG |
| | Rev | GGCCGGGATCCAAGAAACCACATTTTTAAAGTCTTTG |
| GFP-VPS13A (855–1700) | FW | ATGAGCTCGAGGGGGCGGCCTTCAGTTTCCAACTGGAGTTAAA |
| | Rev | GGCCGGGATCCAAGAAACCACATTTTTAAAGTCTTTG |
| GFP-VPS13A (2003–2606) | FW | TTCTGCAGTCGACGGTACTTCAGTCCCACTGTCTGTTTACG |
| | Rev | TCAGTTATCTAGATCCGGTGGGTTAGGCGAACCGGAACATTAGTGTCC |
| GFP-VPS13A (2615–3174) | FW | TTCTGCAGTCGACGGTACTCTGCAGCCGCATGTAATAGC |
| | Rev | TCAGTTATCTAGATCCGGTGTCAGAGGCTCGGAGAAGGTTCTCTTG |
| VPS13A-GFP ΔFFAT | FW | AAGATGATTCAGAGGAGAGTCCCTTGGAAGAACCTC |
| | Rev | GAGGTTCTTCCAAGGGACTCTCCTCTGAATCATCTT |
| GST-VPS13A (2–854/D845A) | FW | TCAGAGGAGGAATTTTTTGCTGCACCATGTAGTCCCTTG |
| | Rev | CAAGGGACTACATGGTGCAGCAAAAAATTCCTCCTCTGA |
| his VAP-A | FW | CCAGCCACATATGATGGCGTCCGCCTCAGGGGCCATG |
| | Rev | GGCAGGAGCGGATCCCTACAAGATGAATTTCCCTAGAAAGAATCC |
| GST-VPS13A (2-835) | FW | ATGCTAGTCGACTGTTTTCGAGTCGGTGGTCGTG |
| | Rev | GGTATAGCGGCCGCACAGATGGAAGTTCCAAGAGAGG |
| GST-VPS13A (2-854) | FW | ATGCTAGTCGACTGTTTTCGAGTCGGTGGTCGTG |
| | Rev | ATTTAAGCGGCCGCAGGTTCTTCCAAGGGACTACATG |
| GST-VPS13A (835–1700) | FW | GCACTGGAGTCGACTTCTGAAGATGATTCAGAGGAG |
| | Rev | CCGGAAGCGGCCGCAAGAAACCACATTTTTAAAGTC |
| FFAT domain only | FW | TCGACTGAATTTTTTGATGCACCATGTGC |
| | Rev | GGCCGCACATGGTGCATCAAAAAATTCAG |
| GFP VAP-A | FW | AGGCCGGAATTCTCAAAATATGATGGCGTCCGCCTCAG |
| | Rev | GCTTCCTTTCGGGCTTTG |
| mCherry VAP-A | FW | ATTGGCTCCGGATATATGATGGCGTCCGCCTCAG |
| | Rev | ATTCCGCTCGAGGCTTCCTTTCGGGCTTTG |

## Generation of VPS13A KO cell line

Human *VPS13A* gene was targeted via a CRISPR/Cas9 approach (*Ran et al., 2013*) using guide sequence GACGTGTTGAACCGGTTCTT in exon 1, positions + 25 to+44 of *VPS13A* cDNA. Oligonucleotides HsA01-01t-F (5'-CACCGACGTGTTGAACCGGTTCTT-3') and HsA01-01t-R (5'-AAACAA-GAACCGGTTCAACACGTC-3') were annealed and used to replace the *Bbs*I 22nt-fragment in the single-guide RNA (sgRNA) sequence of plasmid pSpCas9(BB)−2A-Puro (PX459), also expressing the *Streptococcus pyogenes* Cas9 nuclease and the puromycin resistance gene, to obtain the targeting plasmid 48139-HsA01-01t. MRC5 cells (150,000 per well in 6-well plate) were grown in complete medium as described above and transfected with the targeting plasmid (2.5 µg per well) using TurboFect transfection reagent (Fermentas) (2 µl per µg DNA) according to manufacturer's instructions. Puromycin selection (3 µg / ml) was applied for two days, starting one day after transfection. The remaining cells were then washed, collected by trypsinization, diluted and seeded in 15 cm culture dishes with complete medium without puromycin. Colonies were picked after two weeks and cells were expanded in normal growing conditions. To detect *VPS13A* knock-out clones, these cells were characterized by Western blotting (see *Figure 5—figure supplement 1–2*) to analyse the expression levels of endogenous VPS13A protein. Clone #4, (full name: MRC5-SV2_A01-01t_A2b), with no detectable VPS13A signal, was selected for further experiments.

## Immunoblotting

Fly heads were processed as described before (*Vonk et al., 2017*). Cells were homogenized by sonication in 2x Laemmli buffer containing urea (Sigma) and DTT (Sigma) to a final concentration of 0.8M and 50 mM respectively. Afterwards the homogenates were boiled at 99°C for 5 min. The indicated proteins were separated with 8% polyacrylamide gels and overnight wet transfer, or on 10% or 12% mini protean TGX stain-free gels (Bio-Rad). Stain-free gels were activated and imaged with the ChemiDoc imager (Bio-Rad) before transfer to PVDF membranes using the Trans Blot Turbo System (Bio-Rad). The membranes were blocked in 5% fat free milk for 1 hr at room temperature and rinsed in PBS-Tween 20. Incubations with primary antibodies were done overnight at 4°C followed by incubations with secondary antibodies for 1.5 hr at room temperature. The following primary antibodies were used: anti-ATP5A (Abcam, 1:1000), anti-DRP1 (to detect total DRP) Cell Signaling 1:500), anti-p (hospho)DRP1 ser616 (Cell signaling, 1:1000), anti-GAPDH (Fitzgerald 1:10,000), anti-GFP (Clontech 1:1000), anti-GST (Santacruz Biotechnology, 1:1000), anti-EGFR (Santacruz Biotechnology, 1:1000), anti-LAMP1 (Abcam, 1:1000), anti-Myc (Enzo Life Sciences, 1:1000), anti-PLIN2 (Abcam, 1:1000), anti-Rab7 (Abcam, 1:1000), anti-TOMM20 (BD biosciences 1:1000), anti-α tubulin (Sigma, 1:5000), anti-VAP-A (Santa Cruz Biotechnology, 1:1000), anti-VAP-B (Sigma, 1:1000), anti-VPS13A (Sigma,1:1000), anti-VPS13A (H-102) (Santa Cruz Biotechnology, 1:500), *Drosophila* VPS13A #62 (*Vonk et al., 2017*), VPS13C (Sigma). Membranes were developed using ECL reagent (Thermo Fisher Scientific) and the signal was visualized using the ChemiDoc imager (Biorad), images exported as. tiff files and densitometric analysis of band intensities was performed using ImageJ software.

## Immunofluorescence

For fixed samples, cell were seeded in Poly-L-Lysine coated (Sigma-Aldrich) cover slips and allowed to settle for 24 hr. Afterwards the cells were fixed with 3.7% formaldehyde or 4% paraformaldehyde (Sigma Aldrich) for 20 min, washed briefly with phosphate-buffered saline (PBS) + 0.1% Triton-X-100 (Sigma Aldrich) and permeabilized with PBS + 0.2% Triton-X-100. The slides with cells were then incubated with primary antibody (anti-TOMM20, 1:200; anti-Myc, 1:500; anti-FLAG, 1:500) at 4°C overnight and after an additional washing step in PBS + 0.1% Triton-X-100 probed with matching secondary antibodies (Molecular Probes) for two hours at room temperature (RT). The cell nucleus was detected by DAPI staining (0.2 µg/ml) (Thermo Fischer Scientific). Finally the samples were mounted in 80% glycerol and analysed with one of the confocal microscopes listed below.

For LipidTox staining, cells were fixed as described above, then quenched for 10 min in 50 mM $NH_4Cl$ in PBS, permeabilized for 5 min with 0.1% Triton x-100 in PBS followed by incubation with LipidTox dye (Thermo Fischer Scientific 1:200) for 30 min at room temperature. Cells were mounted using citifluor mounting medium (Agar Scientific) and imaged immediately. Mitotracker (Thermo

Fischer Scientific) was added for 20 min in serum free medium at a concentration of 100 nM (for live) and 200 nM (for fixed) cells, after which the cells were fixed and co-stained as described above.

For Live imaging procedures, cells were seeded in 35 mm glass bottom dishes coated with poly-D-lysine (Mat Tek). BODIPY-FA (Thermo Fischer Scientific) 1 μM was added for 30 min (live). Live cell recordings (600 ms/frame) were made using a DeltaVision confocal microscope. Prior to imaging, the cage was allowed to reach 37°C and cells were supplemented with 5% $CO_2$ throughout the entire recording. Images were deconvoluted by the SoftwoRx software and stored as movies.

## Cytosol and membrane fractionation

Around 4–5, 90% confluent, T75 flasks of HeLa cells were scraped in ice cold PBS and resuspended in 1 ml homogenization buffer HB (50 mM Tris HCl pH 7.5, 150 mM NaCl, 1 mM EDTA, Protease inhibitor cocktail (Roche)). The cell suspension was lysed through 2 freeze-thaw cycles and 20 strokes using a 27 gauge needle. The nuclei and intact cells were pelleted by centrifugation for 5 min at 800 g, and the resulting postnuclear supernatant (PNS) was applied to ultracentrifugation at 100,000 x g, for 1 hr, using a TLA 100.3 rotor in a Beckman Coulter, to generate the cytosol and the membrane fraction. The membrane fraction was washed in 1 ml of HB and centrifuged 1 hr at 100,000 x g. All centrifuge steps were carried out at 4 degrees. Laemmli sample buffer was added to the cytosol and membrane fractions, samples were quantified and 20 μg of proteins of each sample were loaded on SDS-gel and processed for Western blot analysis.

## Digitonin based subcelullar fractionation

Digitonin extraction of cytosolic proteins was performed according to (*Holden and Horton, 2009*). Briefly, HEK293T cells were cultured in 5 cm dishes. When about 70% confluent, cells were collected by trypsinization, washed with ice cold PBS and resuspended in 5 ml of digitonin buffer (150 mM NaCl, 50 mM HEPES PH = 7.4, 25 ug/ml digitonin, protease inhibitor cocktail (Roche)). After rolling the suspension for 10 min at 4 degrees, the tube was centrifuged at 2000 x g for 5 min. The supernatant was collected as cytosolic fraction. The pellet was washed once with cold PBS and resuspended in 5 ml of NP-40 buffer (150 mM NaCl, 50 mM HEPES PH = 7.4, 1% NP-40, protease inhibitor cocktail (Roche)). After rolling the suspension for 30 min at 4 degrees, the tube was centrifuged at 7000 x g for 5 min. The supernatant was collected as membrane fraction. All centrifuge steps were carried out at 4 degrees. Both the cytosolic and membrane fractions underwent TCA precipitation and equal amounts of proteins were processed for immunoblotting as described above.

## Membrane extraction

The membrane fractions (after digitonin extraction) were resuspended in HB (control), 1M KCl, 0.2M sodium carbonate (pH 11) or 6M urea for 45 min shaking on ice, and then centrifuged at 4 degrees, 100,000 x g for 1 hr, using a TLA 100.3 rotor in a Beckman Coulter, obtaining soluble (supernatant) and insoluble (pellet) fractions. Laemmli sample buffer was added to the insoluble and soluble fractions, samples were quantified and 20 μg of proteins of each sample were loaded on SDS-gel and processed for Western blot analysis.

## Subcellular fractionation

For subcellular fractionation around 5–6, 90% confluent, T75 flasks of HeLa cells were scraped in ice cold PBS and resuspended in 1 ml of homogenization buffer HB1 (50 mM Tris HCl pH 7,5, 150 mM NaCl, 1 mM EDTA, Protease inhibitor, 0.25 M sucrose). The cell suspension was homogenized as previously described (see cytosol and membrane fractionation) to obtain PNS. The PNS was then loaded onto a 10 ml continuous sucrose gradient containing 5–55% (w/v) in HB1, and was spun at 4 degrees at 274, 000 x g for 4 hr using a swinging bucket SW41 rotor in a Sorvall Discovery 90se. Gradient fractions of 0.5 ml were collected from top to bottom. The proteins in each fraction were concentrated using trichloroacetic acid (TCA) precipitation and resuspended in 75–100 μl of sample buffer. All the procedures were performed on ice. Equal volume of each fraction was loaded onto SDS-gel and processed for Western blot analysis.

## Immunoprecipitation

HeLa cells were washed once with ice cold PBS and then scrapped into ice cold PBS. After centrifugation the cells were resuspended in immunoprecipitation buffer (IB) (50 mM Tris HCl, 150 mM NaCl, 1 mM EDTA, 1.5 mM MgCl$_2$, 10 mM KCl, 1% Triton X-100, pH 7.6) supplemented with protease inhibitor cocktail (Roche). Cells were then snap frozen in liquid nitrogen twice and in between passed through a 26 gauge needle, 10 strokes. The resulting homogenate was spun down at 10,000 x g for 10 min, the supernatant was recovered and subjected to overnight immunoprecipitation using indicated antibodies or control IgG of the same host. Immunoprecipitates were enriched on agarose beads (Santa Cruz) at 4 degrees for 1.5 hr. Protein A/G plus agarose beads (Santa Cruz) were gently washed with IB and resuspended in 2x Laemmli buffer containing DTT and urea and processed for immunoblotting as described above. Co-immunoprecipitation using GFP-Trap beads (Chromo Tek) was done according to manufactures instructions.

## In vitro protein-protein interaction

GST-tagged protein coding plasmids were transformed in E.coli BL21 and bacteria was grown overnight in 1 liter Luria Broth (LB) medium. When the bacteria suspension reach the OD600 of 0.6 protein expression was induced using IPTG 1 mM for 4 hr. Cells were pelleted by centrifugation at 5000 x g for 15 min and lysed by sonication in 40 ml lysis buffer (LB) (50 mM Tris HCl, PH +7.5, 150 mM NaCl, 5% glycerol, 0.1% Triton S-100, 1 mM PSMF, protease inhibitor cocktail (Roche)). Debris was removed by centrifugation at 4000 x g for 15 min and the clean supernatant was mixed with 1 ml glutathione beads (GE healthcare), incubated for 2 hr at 4 degrees. Beads were washed with LB 3 times. For protein-protein interaction assays, a bacterial lysate that contains His-VAP-A or HeLa cell lysate was added to the GST-VPS13A enriched beads and incubated at 4 degrees overnight. Beads were gently washed with LB and resuspended in 2x laemmli buffer containing DTT and urea, incubated for 5 min at 99°C and processed for immunoblotting as decribed above.

## Splics

Cells were transfected as described (*Cieri et al., 2018*). Briefly, 48 hr after transfection cells were fixed as described above and stained with DAPI. A z-stack covering the cell was acquired using a Leica SP8 confocal microscope. Z-stacks were processed using ImageJ with the VolumeJ plugin (http://bij.isi.uu.nl/vr.htm). The image was then used to count ER-mitochondria contact sites.

## Mitochondria morphology

Mitochondria were scored according to (*Rambold et al., 2011*). Briefly, 80% confluent cells were washed once with HBSS and incubated for 16 hr in HBSS or normal medium at 37°C + 5% CO$_2$. The cells were stained with TOMM20 as describe above and mitochondrial morphology was scored in three types as follows and partly based on (*Rambold et al., 2011*) : type 1; fragmented, mainly small and round mitochondria, mainly localized and densely packed at one site of the nucleus; type 2; intermediate, mixture of round densely packed and shorter tubulated mitochondria more surrounding the nucleus; and type 3; tubulated dispersed and long mitochondria..

## Mitophagy

Mitophagy was performed as previously described (*Narendra et al., 2008*). Briefly, MRC5 control and clone 4 cells were transfected with FLAG-parkin. 24 hr post transfection the cells were treated with 20 µM CCCP (Sigma) or DMSO for 48 hr. Cells were labelled with anti-TOMM20, anti-FLAG and stained with DAPI. To correct for transfection efficiency, the TOMM20 mean value was measured exclusively in Parkin-positive cells using ImageJ.

## LD fractionation

HeLa cells were collected by trypsinization and washed once with PBS. After centrifugation, cell pellets were resuspended in detergent free homogenizing buffer (50 mM Tris HCl, 150 mM NaCl, 1 mM EDTA, 1.5 mM MgCl$_2$, 10 mM KCl, PH 7.6) supplemented with protease inhibitor cocktail. Cells were snap frozen in liquid nitrogen 3 times and in between passed through a 26 gauge needle 20 strokes. Nucleus and unbroken cells were removed by spinning down at 1600 x g for 5 min. The supernatant was recovered and mixed with equal volumes of 0.25M sucrose in homogenizing buffer.

After saving an input, the sample was loaded on top of a discontinuous sucrose gradient prepared by layering 1 ml of 30%, 20%, 10% and 5% sucrose in SW55 ultracentrifuge tube. The gradient was centrifuged at 4 degrees for 3 hr at 194,000 x g, using an ultracentrifuge in a Sorvall discovery 90se. The tubes were carefully removed and 8 fractions of 600 µl were collected from top to bottom. 600 µl of the top fraction containing LDs were collected using a 20 µl pipette with a tip cut off. The refractive index of each fraction was measured and correlated to the linearity of the sucrose concentration throughout the tube. The bottom part containing the pellet was resuspended with buffer to a final volume of 600 µl and was neither included in the refractive index measurement nor in the quantification of protein distribution among gradients. Proteins from each fraction were precipitated using the TCA precipitation method. Equal amounts of proteins were processed for immunoblotting as described above. The amount of protein in each fraction was calculated as a ratio of the densitometric signal in each fraction to the sum of the total protein in fractions 1–8 (Protein per fraction (%) =densitometric signal of a fraction/sum of total densitometric signal (1-8) x 100)

## FACS analysis

Cells were collected by trypsinization, centrifuged and washed and resuspended in 200 µL PBS with Nile Red stock solution diluted to 1:500. Cells were incubated at room temperature for 15 min. After incubation the cells were washed with PBS and resuspended in 300 µl PBS. Finally, the cells were measured on a FACScalibur (BD) and analyzed with FlowJO V10. For this experiment the mean fluorescence intensity of ~10,000 cells was analyzed.

## Fly stocks and genetics

Fly stocks were maintained and experiments were done at 25°C on standard agar food unless indicated otherwise. The *Vps13{PB}c03628* stock (here referred to as *Vps13*) was acquired from the Exelixis stock centre and isogenized to the *w1118* stock (*Vonk et al., 2017*). The *UAS-hVPS13A* expressing *Drosophila* line in the *Vps13* mutant background has previously been described in *Vonk et al., 2017*. For the rescue experiment two stocks were created, *Vps13/CyO; Act-Gal4/TM6B* and *Vps13/CyO, UAS-hVPS13A,* and mated to produce the offspring listed in *Figure 9* (homozygous or heterozygous *Vps13* mutant flies expressing or not expressing *UAS-hVps13A* ubiquitously under the control of *Act-Gal4*).

## Whole mount staining of fly retinas

LD staining of adult fly retinas was performed as described previously (*Liu et al., 2015*). Images of male flies are shown (*Figure 9*), female flies were also stained and showed a comparable phenotype.

## Mitochondria membrane fractionation

Fractionation studies were performed as described previously (*Sugiura et al., 2017*; *Mattie et al., 2018*). Briefly, cells were seeded in 10 × 14,5 cm dishes and when 90% confluent, were scraped into ice cold PBS after a wash step with PBS. After centrifugation they were resuspended in 5 ml homogenization buffer HB2 (30 mM Tris-HCL pH 7.4, 225 mM mannitol, 75 mM sucrose). The cell suspension was then lysed using a 27 gauge needle (20 strokes). Afterwards it was centrifuged at 1000 x g for 10 min, transferred to a new tube an spun again at 1000 x g for 5 min. The postnuclear supernatant (PNS) was transferred and spun for 15 min at 8000 x g, the cytosolic and membrane fraction at 100,000 x g for 30 min, using a TLA 100.3 rotor in a Beckman Coulter Centrifuge, to separate the two fractions. The membrane fraction was then washed with HB2 and centrifuge 100,000 x g for 30 min.

After centrifugation the pellet for the PNS was resuspended in mitochondria resuspending buffer MRB (250 mM mannitol, 5 mM HEPES pH 7.4, 0,5 mM EGTA) and passed through a 27 gauge needle once. It was centrifuged for 15 min at 8000 x g then the mitochondria resuspended in 1 ml MRB. Half of the sample was usedfor proteinase K treatment and the other half for the alkaline carbonate extraction (see below).

For Proteinase K treatment different concentrations (as indicated in Fig X) of proteinase K (Fermentas) were added to the mitochondria either with or without 2% Triton X-100 and incubated on ice for 30 min.

For Alkaline carbonate extraction the mitochondria were centrifuged at 8000 x g for 5 min, resuspended in 0,1 M $Na_2CO_3$. They were then incubated for 30 min on ice, vortexed every 10 min and finally centrifuged at 100,000 x g for 30 min. All centrifuges steps were done at 4 degrees. Proteins from each fraction were precipitated using the TCA precipitation method. Equal amounts of proteins were processed for immunoblotting as described above.

## Hardware/software used for imaging/image work

Confocal images were collected by

- DeltaVision confocal microscope (Applied Precision) fitted with 60x or 100x oil immersion objective. Images from the Delta Vision microscope were deconvolved by the SoftwoRx software (Applied Precision) and saved as movies or exported as. tiff files using ImageJ (NIH). The Delta Vision was used for: *Figure 2* (A-C/E), 3 (D), 4 (E/E'), 6 (A-C), 7 (D/D'), 8 (E/E'), S3 (D), S4 (B/B').
- Leica SP8 confocal laser scanning microscope fitted with a 63x oil immersion objective and images were exported as. tiff files using the Leica software. The Leica SP8 was used for: *Figure 3* (A/B), 5 (A/B/C'), 8 (A/A'), S1 (C-F'), S3 (B-C).
- Zeiss 780 NLO confocal microscope fitted with a 40x oil immersion objective +optical zoom. Zeiss Zen software was used to capture the images and export them as. tiff files. The Zeiss 780 was used for: *Figures 3 (C)* and *9* (A-D''').

ImageJ (NIH) was used for quantifying LDs (*Figure 8B and D*), mean gray intensity (*Figure 5D*), line scan (*Figure 2C", E and 6C'*), SPLICS with volumeJ plugin (*Figure 5A', B'*), lipid movement (*Figure 8F,H*) and colocalization with JACoP plugin (*Figure 4—figure supplement 3*).

Adobe Photoshop and Illustrator (Adobe Systems Incorporated, San Jose, California, USA) were used for image manipulation (changing intensity and cropping of images) and image assembly as well as creating the schematics.

## Statistical analysis

All experiments are presented as mean of at least three independent experiments ± SEM, unless stated otherwise in the legends. Statistical significance was determined by a two-tailed unpaired Student's t test where applicable. Statistical P values $\leq$ 0.05 were considered significant (*p$\leq$0.05, **p$\leq$0.01, ***p$\leq$0.001). Data were analysed using GraphPad Prism (GraphPad Software, San Diego, CA, USA)

## Acknowledgments

We are grateful to F Reggiori and C Rabouille for their critical reading and helpful discussions. The authors acknowledge support from Wellcome Trust (090532/Z/09/Z) and Advocacy for Neuroacanthocytosis Patients to AV-B and APM, support from NWO (VICI 865.10.012) to OCMS.

## Additional information

### Funding

| Funder | Grant reference number | Author |
|---|---|---|
| Advocacy for Neuroacanthocytosis Patients | | Anthony P Monaco<br>Antonio Velayos-Baeza |
| Wellcome | 090532/Z/09/Z | Antonio Velayos-Baeza |
| Nederlandse Organisatie voor Wetenschappelijk Onderzoek | 865.10.012 | Ody CM Sibon |

The funders had no role in study design, data collection and interpretation, or the decision to submit the work for publication.

## Author contributions
Wondwossen M Yeshaw, Conceptualization, Data curation, Formal analysis, Funding acquisition, Investigation, Methodology, Writing—original draft, Writing—review and editing; Marianne van der Zwaag, Conceptualization, Data curation, Formal analysis, Investigation, Methodology, Writing— original draft, Writing—review and editing; Francesco Pinto, Liza L Lahaye, Data curation, Formal analysis, Investigation, Methodology, Writing—review and editing; Anita IE Faber, Rubén Gómez-Sánchez, Amalia M Dolga, Data curation, Investigation, Methodology; Conor Poland, Sven CD van IJzendoorn, Conceptualization, Investigation, Methodology, Writing—review and editing; Anthony P Monaco, Nicola A Grzeschik, Conceptualization, Data curation, Investigation, Visualization, Methodology, Writing—original draft, Writing—review and editing; Antonio Velayos-Baeza, Conceptualization, Data curation, Funding acquisition, Investigation, Methodology, Writing—review and editing; Ody CM Sibon, Conceptualization, Data curation, Funding acquisition, Investigation, Writing—original draft, Writing—review and editing

## Author ORCIDs
Wondwossen M Yeshaw (iD) http://orcid.org/0000-0002-3134-3458
Rubén Gómez-Sánchez (iD) http://orcid.org/0000-0002-8274-3259
Antonio Velayos-Baeza (iD) http://orcid.org/0000-0002-7717-4477
Ody CM Sibon (iD) http://orcid.org/0000-0002-6836-6063

## Decision letter and Author response
Decision letter https://doi.org/10.7554/eLife.43561.037
Author response https://doi.org/10.7554/eLife.43561.038

## Additional files

### Supplementary files
• Transparent reporting form
DOI: https://doi.org/10.7554/eLife.43561.036

### Data availability
All data generated or analysed during this study are included in the manuscript and supporting files.

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
