## [Decision Letter]

[Editors’ note: a previous version of this study was rejected after peer review, but the authors submitted for reconsideration. The first decision letter after peer review is shown below.]

Thank you for submitting your work entitled "Human VPS13A is associated with multiple organelles and required for lipid droplet homeostasis" for consideration by *eLife*. Your article has been reviewed by three peer reviewers, and the evaluation has been overseen by a Reviewing Editor and a Senior Editor. The following individuals involved in review of your submission have agreed to reveal their identity: Hugo J Bellen (Reviewer #1). The other reviewers remain anonymous.

Our decision has been reached after consultation between the reviewers. Based on these discussions and the individual reviews below, we regret to inform you that your work will not be considered further for publication in *eLife*.

As you will see from the original reviews, the reviewers found your findings interesting but not fully developed in terms of mechanisms and not fully convincing. One of the reviewers was more enthusiastic about this work, especially the part focused on cell biology, that seems to be closer to be worked out than the organismal part. Overall, there is a good agreement that in the current version the story is not ready to be revised for *eLife*.

*Reviewer #1:*

In this manuscript, Yeshaw et al., explore the subcellular localization and function of VPS13A in vivo and in vitro using mammalian cells. They show that VPS13A is a membrane associated protein that also associates with VAPA via its FFAT domain and that it responds to metabolic stimuli by altering its localization from mitochondria to lipid droplets. Loss of VPS13 also induces LD formation. They also present data that indicate LD accumulation in glial cells in flies in which the sole VPS13 gene is knocked down. Though the findings presented here are interesting they do not rise to a paper for *eLife*. The data raise numerous questions that remain unanswered and the authors do not try to tackle the molecular mechanisms that may underlie LD accumulations

1) Why are glia affected and not neurons or other cells in flies whereas the authors study show LD accumulations/increase in non glial cells in mammalian cells when Vps 13 is decreased? How do they tie these observations? Also, how do they know that the cells that accumulate LD are glia? Are they reminiscent of glia, as stated? The TEM quality does not seem to establish that these are glia. They can look in the eye where the glial cells are easy to identify (pigment cells). Why is LD production enhanced when Vps 13 is lost? What pathway is affected? *JNK*? Is ROS elevated?

2) What is the interplay between the ER, Vps13 and LD formation?

3) What happens when Vap A is lost? To my knowledge LD have never been documented when VapA or B or both are lost or overexpressed. Has this been missed in the past? How does this tie in with their observations? Is VapB not involved? VapA and B are typically redundant.

4) What are the proposed molecular mechanisms that underlie the observed degeneration? Is the demise of neurons related to the LD accumulation in glia or is it independent? Is this a primary or secondary defect? Is there elevated ROS that causes the production of peroxidated lipids and LD formation when Vps 13 is lost? Is mitochondrial function impaired? Complex 1/complex 3?

I understand that the authors cannot answer all these questions but there is no attempt to explore any of these avenues and much of the data remain therefore descriptive.

Specific comments:

The switching back and forth between two metabolically different cell lines makes the data difficult to interpret and the images are not on par with other publications that examine ER mitochondria contact sites.

In the Introduction, the authors noted that "the cellular localization and function of VPS13A is largely unknown". However, VPS13A has been identified via mass spec to be localized and associated with the mitochondria (https://www.ncbi.nlm.nih.gov/pubmed/28441135), lipid droplets (https://www.ncbi.nlm.nih.gov/pubmed/21870882) and ER in mammalian cells (human and others).

In Figure 2D, no portion of the VAP13A protein is shown to overlap with sec61B. However, in Figure 3B, full length VPS13A colocalizes with sec61B. Can the authors speculate ?

Figures 3 B and C are not of publication quality and it is impossible to tell whether VPS13A is at the ER-mitochondrial contact sites or just present in both organelles. Also, another panel showing the overlap of BFP-sec61B and mitotracker is necessary to show that these two do not colocalize.

Colocalization analysis (as done in Figure 2B') would be useful for all of these subsequent images where only a single image is shown. Moreover, a quantification of colocalization based on many cells would further support the author's claim.

In comparing Figure 5A and 5B, VPS13A is larger in 5B – are these images taken at the same magnifications? Moreover, Nile Red is fluorescent in both the red and green channels and should not be used in conjunction with any imaging in the 488 channel. The authors should use BODIPY495 instead.

In Figure 5A and 5B, 100% of VPS13A is not found as a ring (as in Figure 2B) but shown to be colocalized with lipid droplet markers. Are these images taken as a stack as opposed to a single slice? What accounts for this difference? Moreover, what percentage is a small percentage ? and are these cells undergoing cell death?

In comparing Figure 6G and 7A, where two different cell types are used to study the relationship between VPS13A and lipid droplets, supplementing HEK cells with oleate leads to less LD compared to un-supplemented U20S cells with a mock transfection. This metabolic difference between these cells raises the question whether conclusions can be drawn about VPS13A localization and function in relation to metabolic changes.

For all quantifications, please provide the number of cells quantified and number of biological replicates.

Is there a loss of mitochondria-ER contact sites in the VPS13A mutant fly brains?

The authors argue that VPS13A is translocating from the mitochondria to LD upon oleate induction. However, since the authors show that VPS13A is also on the ER membrane and LDs originate from the ER membrane, what arguments can the authors provide against an ER origin of VPS13A present on LD?

Finally, an ER association of VPS13A would also explain how LDs are "stabilized" when associated with VPS13A – LDs associated with VPS13A may still be tethered to the ER membrane.

*Reviewer #2:*

This manuscript by Yeshaw et al. reports the investigation of VPS13A. The authors main claims are that this protein localizes to a contact between mitochondria and ER that is modulated by Ca++. Furthermore, they report localization to cytoplasmic lipid droplets (LDs), especially when cells are incubated in medium containing fatty acids. Consistent with this localization the authors claim a phenotype on LDs in cells and *Drosophila* brain.

Despite increasing interest, the molecular functions of the very large Vps13 proteins are still somewhat enigmatic and progress in this area is in principle interesting, also because mutations in different family members are associated with human disease.

In my assessment however, the experimental data presented in this manuscript are not conclusive and ultimately there is very limited solid progress. In particularly, while the authors claim an important function in LD homeostasis, the data presented in the paper suggest a very minor role at best. For these reasons, I do not believe this work is suitable for publication in *eLife*. The most important points upon which this conclusion is based are listed below:

1) Localization by microscopy is entirely based on overexpressed proteins. While it is possible that these experiments report the localization of the endogenous proteins, this is not sufficient as it is easy to see how overexpression would lead to a pool of protein that is mis-localized. The authors provide some fractionation data, but those data are not very clear in terms of the subcellular organelle where Vps13A localizes and by itself would not be sufficient to conclude on the localization of the protein.

2) The argument for regulation of ER-Mitochondrial localization is really based on one experiment of thapsigargin treatment, in which alternative explanations, e.g. due to indirect effects of the treatment, are equally likely as a Ca++ effect.

3) While I believe it’s likely that Vps13A interacts with VAP, the data presented is minimal and most experiments have next to no controls.

4) The authors claim the protein's localization can switch to LDs. How these two different localizatios are achieved and or regulated is unclear.

5) The authors claim VPS13A is important for LD homeostasis. However, the phenotype observed is mild at best and the characterization lacks even the most basic analyses (e.g. lipid content of the cells, localization of other LD proteins). In addition some of the measurements have problems (some of the values for LD sizes seem to be below the resolution of the light microscope). Moreover, in these experiments, the other Vps13 isoforms are not considered; overall this is a very preliminary analyses, and if anything, suggests a minor role in LD function at best.

6) The phenotype in *Drosophila* is interesting but also not comprehensively analyzed; only some EMs are shown. The authors would have to provide at least TG and other lipid measurements, and some evidence (e.g. by IF and light microscopy) that the structures observed are LDs.

*Reviewer #3:*

Yeshaw and colleagues have explored the potential role of VPS13A a member of a small family of 4 related proteins and which is mutated in the neurological disorder Chorea Acanthocytosis. The gene encodes a very large cytosolic/membrane associated protein with conserved Chorein, DUF and ATG domains. Like for other family members, little is known about the exact role of VPS13A. Most knowledge is gathered through KD or KO strategies in different model systems, leading to the conclusion that VPS13A has a multitude of cellular functions. The authors explored here in more detail the localization and subcellular functional dynamics in cellular models. They demonstrate that VPS13A is involved in (and localizes to) membrane contacts between the ER and mitochondria; this interaction is calcium dependent and mediated through interaction of the FFAT domain with VAP-A, which they explored using mutational analysis. They further show that upon fatty acid addition, VPS13A is released from the ER and associates, in a FFAT-domain independent way, with lipid droplets. Functionally, VPS13A appears to affect LD size and its association with LDs slows their motility. The effects of VPS3 deficiency on LD appearance were finally validated in a mutant fly model using EM. Overall, the cell and molecular biology is of high quality and the authors make a major contribution to the potential role(s) of this VPS13 member, albeit the full mechanism and physiological importance with respect to the clinical phenotype in ChAc is not strongly revealed. A major concern remains the validation of the cellular data in mutant flies. The fly has only one VPS13 orthologue while in mammals, four variants are known, all with different functions (for instance the VPS13C has opposite effects on LDs compared to VPS13A) and links to very distinct diseases. It follows that the phenotype of the mutant fly cannot be correlated to the function of a specific variant, while the authors actually do that. A possible way to address this is to rescue the mutant fly with each human VPS13 variant and test to what extent they all rescue the phenotype or only certain features. In the best case, the common function among all VPS13 forms could be identified in this way. After all, to what extent is the observed cellular phenotype (smaller, more mobile LDs) correlated with the mutant fly phenotype (larger LDs)?

Specific comments that should be addressed:

Figure 1C-D: the authors explore the membrane association of VPS13A using chemical agents. It suprises me why they don't use 'golden standard' approaches using bicarbonate, or detergents like TX114-phase partitioning or other that more selectively extract associated proteins.

Figure 2 and Figure 2—figure supplement 1: Truncated forms of VPS13A are used to show that the C-terminus is required for mitochondrial association. The authors refer to Figure 2—figure supplement 1 to state that the C-terminal region localized to the mitochondria in different cell lines: this is an overassumption as Figure 2D and Figure 2—figure supplement 1 only shows a co-staining of an ER marker with truncated VPS13A, not with a mitochondrial marker. Also the localization of the C-terminus is very different between different cell lines (compare A with C for instance: can the authors exclude that in A-B these structures are not other compartments like LDs?). The only evidence is in fact Figure 2B and 2E (triple staining). Small remark: the yellow arrow is not on the right position in panel A, merged inset.

Figure 4E: same remark. Since only a co-staining with an ER-marker, VAP-A, is shown, one cannot conclude from these panels that VPS13AdeltaFFAT is shifted from ER to mitochondrial localization. Given that VPS13A can also locate to LDs, a triple staining or dual staining with a mitotracker and higher resolution (zoomed insets) is needed.

Figure 5C: these data show the relocation of GFP-VPS13A to LDs upon addition of OA. However these seem to be snapshots at different time points from different regions of the coverslip. Hence, the last sentence of this paragraph, 'Live-cell imaging showed that newly formed LDs… gradually acquired VPS13A…', is an overstatement as not the same LD is followed in time. Furthermore, one cannot deduce from these panels nor from the video that GFP-VPS13A goes to newly formed LDs, but instead associates/dissociates from existing LDs. Data in Figure 7E and Video 3 are also not really supporting this. To experimentally test this, probes like LiveDrop should be used as these mark the earliest stages of LD formation from the ER. Furthermore, is GFP-VPS13A recruited to LDs from nearby ER-mito contacts as this would suggest a more intricate association of ER-mito contacts with LD formation?

Figure 6: Using cell fractionation, the authors show that OA induces a shift of VPS13A to the floating LD fraction. The quantification shows clearly a trend but are these increases statistically significant (B-B' and D-D')? This is important since VAP-A also increases indicating at least some contamination? Some antibodies also give multiple bands: specific bands to PLIN2 and LAMP1 should be more clearly indicated to interpret the data. The authors have also used two concentrations of OA (250 and 500µM). While panel 6F shows the dose-dependent increase in VPS13A with LDs (using imaging), this is less obvious for the biochemical (flotation gradients) data. In fact, there is little difference when looking at the relative shifts of VPS13A and PLIN2 between 250 and 500µM.

Figure 7: In some cases, like in this figure the authors switch to U2OS cells. Why? I would also include a zoomed inset for panel D: the shifts in frame 1 vs frame 2 are difficult to discern. Also these images do not really allow to distinguish large from small LDs (they seem in both wt and siRNA to be overall small). Better images are needed that clearly demonstrate the point the authors are making. In addition, instead of measuring% overlap, it might be better to measure 'distance travelled' as a readout for mobility (and thus to show differences). The conclusion of this paragraph is also overstated: 'In the absence of VPS13A' should be 'When VPS13A is downregulated'. Did the authors generate CRISPR/Cas9 KO cells of VPS13A and if so, are the phenotypes worse?

In some of the panels with confocal data, the images seem to be moved or shifted. For instance in Figure 2—figure supplement 1C, the Rab5QL enlarged endosomes have a suspicious double membrane and the Rab7QL endosomes look fuzzy. Another example is Figure 7A where each dot is a little stripe as this is a timelapse (or like a picture of the night sky with long opening time).

In the Discussion section the authors refer to the work of the Bellen lab to state that LDs are formed in glia cells in response to oxidative stress. In the meantime, they published a follow-up paper on a non-cell autonomous mechanism which should be included (Liu et al., 2017) and may help to explain (the differences with) the observed phenotype.

[Editors’ note: what now follows is the decision letter after the authors submitted for further consideration.]

Thank you for resubmitting your work entitled "Human VPS13A is associated with multiple organelles and influences mitochondrial morphology and lipid droplet motility." for further consideration at *eLife*. Your revised article has been favorably evaluated by Anna Akhmanova (Senior Editor), a Reviewing Editor, and three reviewers.

The manuscript has been improved but there are some remaining issues that need to be addressed before acceptance, as outlined below:

Overall, the reviewers are satisfied and they note the value of the paper, also in the light of currently emerging publications on Vps13. However, they agree that some of the aspects still need strengthening. Specifically and most importantly, the reviewers are not yet fully convinced by the data concerning a putative switch from ER-mito to ER-LDs. There is a general agreement that this switch is not sufficiently substantiated. The authors could still address this with their current data by quantifications and providing controls.

Furthermore, in the light of recent publications please make sure to avoid the statements concerning an "unknown function" of Vps13, such as the one in the Introduction.

---

## [Author Response]

[Editors’ note: the author responses to the first round of peer review follow.]

We are happy to resubmit our thoroughly revised manuscript. We have performed all the experiments requested by the reviewers and added new ones to further strengthen the message of our study.

In the final stage of this experimental work (August 2018), we noticed that the Journal of Cell Biology published a manuscript by Kumar et al., “VPS13A and VPS13C are lipid transport proteins differentially localized at ER contact sites”. Although partly overlapping with our present manuscript, this JCB paper confirms exactly the VPS13A localization we presented in our original submission in January 2018, and therefore strengthens it. Furthermore, our original biochemical studies we presented to validate VSP13A subcellular localization are not presented in the JBC manuscript.

In addition, we now also report a clear mitochondrial phenotype observed in a VPS13 knock- out cell line that we have generated by the CRISPR/CAS9 system (as suggested by reviewer 3) and we report a lipid droplet phenotype in the *Drosophila* Vps13 mutant eye (as suggested by reviewer 1 and 2). This phenotype is rescued by expression of human VPS13A in the *Drosophila* mutant background (as suggested by reviewer 3).

Taken together, as mentioned above, there is overlap between the two manuscripts (immunofluorescence localization of VPS13A) and the parts that overlap are perfectly in agreement. Please note that there are also large non-overlapping parts.

Moreover, the manuscript by Kumat et al. and our work together, reveal an emerging role for VPS13A in lipid transfer at membrane contact sites that when defective leads to a mitochondrial and lipid droplet phenotype.

We addressed all comments raised by the reviewers in the revised version.

As reviewer 1 proposed, we used the fly eye, in which glia cells can easily be identified, to investigate a lipid droplet phenotype observed upon Vps13 loss of function. We now demonstrate that in these cells lipid droplet numbers are increased, consistent with our observations in mammalian cells. Following the suggestion of reviewer 3, we now show that this fly phenotype is rescued by overexpression of human VPS13A. This demonstrates a functional conservation of VPS13A in lipid droplet biology between humans and flies. As suggested by reviewer 1, we discuss these findings in light of the VPS13A- associated neurodegenerative disease. Last, because we agree that the electron microscopy data need more investigation, we have removed them from the revised manuscript.

Using SPLICS assays (to determine contact sites between ER and mitochondria) we now show that the VPS13A KO cells, that we generated, show a significant decrease in ER-mitochondria contact sites. This therefore further suggests a role for VPS13A in maintaining these contact sites.

Using the VPS13A KO cells, we now show evidence for impairment of mitophagy and increased mitochondria fragmentation when compared to control cells.

We also show that VPS13A is needed for the mitochondria elongation that normally occurs upon starvation. In VPS13A KO cells mitochondria fail to elongate, and we demonstrate that phosphorylation on Ser616 of Drp1 is increased in VPS13A KO cells, indicating an increased mitochondria fission in line with the phenotype previously mentioned.

Last, we present a model merging our new data with those of the recent JCB manuscript.

Reviewer #1:

*In this manuscript, Yeshaw et al., explore the subcellular localization and function of VPS13A* in vivo *and* in vitro *using mammalian cells. They show that VPS13A is a membrane associated protein that also associates with VAPA via its FFAT domain and that it responds to metabolic stimuli by altering its localization from mitochondria to lipid droplets. Loss of VPS13 also induces LD formation. They also present data that indicate LD accumulation in glial cells in flies in which the sole VPS13 gene is knocked down. Though the findings presented here are interesting they do not rise to a paper for eLife. The data raise numerous questions that remain unanswered and the authors do not try to tackle the molecular mechanisms that may underlie LD accumulations*

In the revised version we were able to answer more questions. This is possible because of the critical questions and comments raised by the reviewers, because of additional experiments we performed and because of two recently published manuscripts in JCB (Kumar et al., 2018 and Bean et al., 2018) while our manuscript was under revision. All together the molecular mechanism of VPS13A is tackled substantially. In addition increased understanding is obtained about the phenotype caused by absence of VPS13A protein from results obtained from a newly generated VPS13A knock-out cell line. We have rewritten our manuscript now in such a manner to make this clear. The changes are visible via track changes.

1) Why are glia affected and not neurons or other cells in flies whereas the authors study show LD accumulations/increase in non glial cells in mammalian cells when Vps 13 is decreased? How do they tie these observations? Also, how do they know that the cells that accumulate LD are glia? Are they reminiscent of glia, as stated? The TEM quality does not seem to establish that these are glia. They can look in the eye where the glial cells are easy to identify (pigment cells).

We agree with this reviewer that the TEM experiments are not explored to its full extend. As suggested, we have now examined a possible phenotype of lipid droplets in *Drosophila* eyes using fluorescent imaging of lipid droplets, visualized by Nile red and imaged with light microscopy as suggested by reviewer 2. For this we used a validated *Drosophila* homozygous Vps13 mutant. *Drosophila* Vps13 is a gene orthologous to human Vps13A and to Vps13C (Velayos-Baeza, A. et al., 2004). In the pigment (glia) cells of the eye an increase in lipid droplets is observed in homozygous Vps13 mutants, which are absent in control flies and in mutant flies upon overexpression of human VPS13A (as reviewer 3 suggested). These results are provided as well as a Western blot showing the expression of human VPS13A and the absence of *Drosophila* Vps13 in *Drosophila* Vps13 mutants. These data are provided in the revised version as Figure 9.

Why is LD production enhanced when Vps 13 is lost? What pathway is affected? JNK? Is ROS elevated?2) What is the interplay between the ER, Vps13 and LD formation?4) What are the proposed molecular mechanisms that underlie the observed degeneration? Is the demise of neurons related to the LD accumulation in glia or is it independent? Is this a primary or secondary defect? Is there elevated ROS that causes the production of peroxidated lipids and LD formation when Vps 13 is lost? Is mitochondrial function impaired? Complex 1/complex 3?I understand that the authors cannot answer all these questions but there is no attempt to explore any of these avenues and much of the data remain therefore descriptive.

In the revised version we provide possible answers to most of these questions in the discussion. The answers came from our newly added data further underscoring a lipid droplet phenotype and demonstrating a mitochondrial phenotype in VPS13A depleted cells. We combine our new data with data presented in two recently published manuscripts about human VPS13A (Kumar et al., 2018) and yeast Vps13 (Bean et al., 2018). We have combined our findings and the findings of the two JCB manuscripts in a model, presented in Figure 10 of the revised version.

We have changed the discussion thoroughly to make this all more clear in a comprehensive manner.

3) What happens when Vap A is lost? To my knowledge LD have never been documented when VapA or B or both are lost or overexpressed. Has this been missed in the past? How does this tie in with their observations? Is VapB not involved? VapA and B are typically redundant.

We are not sure but we assume that reviewer 1 means that it would be of interest to show the link between VAPA/B and lipid droplets? We agree with this, however, we like to stress that we do not show or claim a link between lipid droplets and VAP-A. We present two different observations 1) VPS13A is localized at ER-Mito contact points and 2) VPS13A is localized at lipid droplets. The FFAT domain of VPS13A binds to VAP-A and we now also demonstrate that VPS13A binds to VAP-B as well (Figure 4—figure supplement 3C). This underscores the statement of reviewer 1 that indeed VapA and B are most likely redundant in VPS13A mediated functions. In case the reviewer means that it is of interest to see what happens with the localization of VPS13A in the absence of VAP-A, we would like to refer to one of the results by Kumar et al., 2018 in which the localization of VPS13A was investigated when VPS13A is overexpressed in VAP-A KO cells,. Under these conditions the localization of VPS13A shifts from an ER localization to a more mitochondrial localization pattern. This result is in agreement with our results demonstrating that VPS13A binds to VAP-A and thereby creates a localization in close association with the ER. We refer to these results in the discussion.

Specific comments:The switching back and forth between two metabolically different cell lines makes the data difficult to interpret and the images are not on par with other publications that examine ER mitochondria contact sites.

We used HeLa cells and HEK293 cells for the biochemical studies. We initiated our work with HeLa cells and later also used HEK293 cells. For the biochemical studies we did not see a difference between the two cell lines in the experiments in which we compared the two. For a subtype of localization studies we used HEK293 cells, because in our hands they gave better transfection efficiency compared to HeLa cells. We now left out all siRNA experiments with the U2OS cells and we added instead data with a MCR5 parental cell line (control) and a newly generated MCR5 VPS13A KO cell line. We used U2OS cells for some of the imaging studies, especially for co-localization studies, because in these cells the morphology of the ER and the mitochondria results in clearer pictures. We now clearly indicate in the Materials and methods section, main text and figure legends which cell types were used for which experiments. In addition, we improved all images of all the figures of the main text, except, Figure 7G (lipid droplets association with VPS13A) and Figure 8F (stills from in vivo imaging of lipid droplet movement).

In the Introduction, the authors noted that "the cellular localization and function of VPS13A is largely unknown". However, VPS13A has been identified via mass spec to be localized and associated with the mitochondria (https://www.ncbi.nlm.nih.gov/pubmed/28441135), lipid droplets (https://www.ncbi.nlm.nih.gov/pubmed/21870882) and ER in mammalian cells (human and others).

Our point was that VPS13A was never localized using immunofluorescence. When our manuscript was submitted in 2017 this was the case. Now the situation is different (Kumar et al., 2018). We have rephrased our text to make this clear, we have included these references and we mention that our results are in agreement with the JBC manuscript.

In Figure 2D, no portion of the VAP13A protein is shown to overlap with sec61B. However, in Figure 3B, full length VPS13A colocalizes with sec61B. Can the authors speculate ?

This observation can be explained because overexpression of the full length VPS13A induces increased ER-mitochondria interaction (as demonstrated by Kumar at al., 2018) and as a result an overlap with sec61B can be observed. We now mention in the discussion this result of Kumar et al. The FFAT motif is required, but not enough, for interaction with VAP-A and ER localization (Figure 4E in the revised version). Thus, it seems that full length VPS13A, or at least fragments bigger than those analyzed here (shown in Figure 2), may be required for ER localization (overlapping signal with sec61B).

Figures 3 B and C are not of publication quality and it is impossible to tell whether VPS13A is at the ER-mitochondrial contact sites or just present in both organelles. Also, another panel showing the overlap of BFP-sec61B and mitotracker is necessary to show that these two do not colocalize.Colocalization analysis (as done in Figure 2B') would be useful for all of these subsequent images where only a single image is shown. Moreover, a quantification of colocalization based on many cells would further support the author's claim.

We now have provided better quality images and have indicated more clearly what is co-localizing and what is not co-localizing and what is in close association. In the revised version in Figure 3A we now demonstrate that the VPS13A and VAP-A signal is similar and overlaps, as indicated by the yellow signal in the overlay image. In Figure 3B we show that VPS13A does not 100% co-localize with mitochondria, but the VPS13A signal is closely associated with the boundaries of the mitochondrial marker mitotracker. Figure 3B shows that VPS13A is not present in mitochondria, but localized to the periphery of mitochondria. in vivo imaging results (Video 2 and Figure 3D) show that VPS13A is not present in the ER but is localized in close association with the ER. Our localization studies are verified by the newly added mitochondrial fractionation data (Supplementary Figure 2). VPS13A also partly co-localizes with BFP-Sec61B, the overlapping signal is indicated with white arrowheads in Figure 3B. We now also show, as the reviewer requests, that BFP-Sec61B does not co-localize with Mitotracker Red, but is detected at some spots in close association with Mitotracker Red. Those localizations are positive for VPS13A as well, indicated with white arrowheads in Figure 3B. Our localization studies are in agreement with Kumar et al., 2018.

In comparing Figure 5A and 5B, VPS13A is larger in 5B – are these images taken at the same magnifications? Moreover, Nile Red is fluorescent in both the red and green channels and should not be used in conjunction with any imaging in the 488 channel. The authors should use BODIPY495 instead.

The Nile red image is removed and LipidTox is now used for all the LD stainings, except for FACS analysis.

In Figure 5A and 5B, 100% of VPS13A is not found as a ring (as in Figure 2B) but shown to be colocalized with lipid droplet markers. Are these images taken as a stack as opposed to a single slice? What accounts for this difference? Moreover, what percentage is a small percentage ? and are these cells undergoing cell death?

This difference is due to a difference in magnification used and the size of the lipid droplets under investigation. We would like to note that the original Figure 2 is a still from a live cell while 5A and 5B (now removed) were fixed preps. We have made this now clearer in the revised manuscript. When lipid droplets are small, the VPS13A ring-like structure is not visible, appearing more like dots, but when lipid droplets are large the VPS13A positive ring-like structure can be visualized. This can be seen in Figure 6B at 120’ after OA addition. Some lipid droplets are large and show a ring-like positive VPS13A signal. This can be visualized upon zooming in our original images (although maybe not in the merged pdf file created after submission). This is now better explained in the text. The small percentage has been rephrased now as follows:” In addition to a localization at areas were mitochondria and the ER are in close proximity we observed that VPS13A is also observed in a punctate and vesicular-shaped pattern.

These vesicular-like structures did not represent mitochondria (Video 1).” Upon close examination, most cells seem to contain small numbers of lipid droplets which, upon expression of full length VPS13A-GFP, are positive for GFP as well. Under control culturing conditions, HEK293 cells seem to only contain a few (1-2) lipid droplets per cell and these are not visible in every focal plane of the images. Therefore the reticular structure (mitochondria-ER contact sites) is usually always captured in an image when full length VPS13A-GFP is visualized, but the same is not true for the ring-like structures (lipid droplets).

In comparing Figure 6G and 7A, where two different cell types are used to study the relationship between VPS13A and lipid droplets, supplementing HEK cells with oleate leads to less LD compared to un-supplemented U20S cells with a mock transfection. This metabolic difference between these cells raises the question whether conclusions can be drawn about VPS13A localization and function in relation to metabolic changes.For all quantifications, please provide the number of cells quantified and number of biological replicates.

The U2OS cells are replaced with MRC5 cells, which we also used as the parental cell line to create a VPS13A KO by CRISPR/Cas9. Now, we do not compare different cell lines in the revised version. We have the numbers of cells for this quantification and will include in the legend. We have removed the quantification of the amount of lipid droplets after oleic acid (OA) in the various cells. We only compare the amount of lipid droplets under normal conditions in control MCR5 cells and in the MCR5 VPS13A KO cells and we found an increase in lipid droplets in the KO. We use OA to induce lipid droplet formation and we demonstrate that, under those conditions of increased lipid droplet formation, VPS13A is more enriched in the lipid droplet fraction. In a separate file we added for each experiment the number of cells quantified and the biological replicates. Upon request we can add these data to the Figure legends.

Is there a loss of mitochondria-ER contact sites in the VPS13A mutant fly brains?The authors argue that VPS13A is translocating from the mitochondria to LD upon oleate induction. However, since the authors show that VPS13A is also on the ER membrane and LDs originate from the ER membrane, what arguments can the authors provide against an ER origin of VPS13A present on LD?

We have now rephrased this argument. We observe in cells that the VPS13A is localized in a reticular pattern and is residing at places where mitochondria and ER are in close contact. VPS13A is also localized at the surface of the scarcely found lipid droplets under control culturing conditions. When OA is added to the medium, lipid droplets are present in increasing amounts and we observed that VPS13A is localized pronouncedly at the surface of these lipid droplets. Biochemical experiments under these altered conditions demonstrate that VPS13A is getting enriched in the lipid droplet fraction. We do not state in the revised version that relocation of VPS13A from ER-mitochondria to lipid droplets does occur, but in the discussion we argue that this may occur. In combination now with the manuscripts in JCB by Kumar et al. and by Bean et al., we favor an adjusted model consistent with the observations of our data and these 2 recently published VPS13-related manuscripts. This is explained and discussed extensively in the discussion.

Finally, an ER association of VPS13A would also explain how LDs are "stabilized" when associated with VPS13A – LDs associated with VPS13A may still be tethered to the ER membrane.

This is indeed possible and demonstrated by the recent report by Kumar et al., where VPS13A is shown to be present at the ER-LD contact sites. This indeed may explain our observation that LDs move less when VPS13A is present because of their attachment via VPS13A to the ER. All this is now combined in the discussion and in our revised model (Figure 10).

Reviewer #2:

This manuscript by Yeshaw et al. reports the investigation of VPS13A. The authors main claims are that this protein localizes to a contact between mitochondria and ER that is modulated by Ca++. Furthermore, they report localization to cytoplasmic lipid droplets (LDs), especially when cells are incubated in medium containing fatty acids. Consistent with this localization the authors claim a phenotype on LDs in cells and Drosophila brain.Despite increasing interest, the molecular functions of the very large Vps13 proteins are still somewhat enigmatic and progress in this area is in principle interesting, also because mutations in different family members are associated with human disease.In my assessment however, the experimental data presented in this manuscript are not conclusive and ultimately there is very limited solid progress. In particularly, while the authors claim an important function in LD homeostasis, the data presented in the paper suggest a very minor role at best. For these reasons, I do not believe this work is suitable for publication in eLife. The most important points upon which this conclusion is based are listed below:1) Localization by microscopy is entirely based on overexpressed proteins. While it is possible that these experiments report the localization of the endogenous proteins, this is not sufficient as it is easy to see how overexpression would lead to a pool of protein that is mis-localized. The authors provide some fractionation data, but those data are not very clear in terms of the subcellular organelle where Vps13A localizes and by itself would not be sufficient to conclude on the localization of the protein.

In the revised version new data is included to show endogenous VPS13A under control culturing conditions is peripherally attached to mitochondria (Figure 2, Figure 3—figure supplement 3). We do believe that the binding of VPS13A via its FFAT domain to VAP-A and VAP-B is convincing. We also now include experiments with a VPS13AΔFFAT construct, which does not co-immunoprecipitate VAP-A and VAP-B (Figure 4F, Supplementary Figure 4C). In addition the SPLICS data (Figure 5) show an involvement of VPS13A in contact sites between ER and mitochondria. Our data are in agreement with the recent manuscript by Kumar et al. in which also overexpression studies were used as well as endogenous- tagged VPS13A for localization studies. Together with our localization data and our biochemical experiments we feel there is now substantial evidence for our reported subcellular localization of VPS13A at ER-mitochondria contact sites and at lipid droplets.

2) The argument for regulation of ER-Mitochondrial localization is really based on one experiment of thapsigargin treatment, in which alternative explanations, e.g. due to indirect effects of the treatment, are equally likely as a Ca++ effect.

We also now have added experiments in which VPS13A depletion is inducing a decrease in ER- mitochondria contact sites by using split-GFP-based contact site sensor (SPLICS) engineered to fluoresce when organelles are in proximity (Figure 5). Kumar et al., 2018 demonstrated that overexpression of VPS13A induces an increase in ER-mitochondria contact sites. Together these data suggest that VPS13A is indeed able to influence ER-mitochondria contact sites.”

3) While I believe it’s likely that Vps13A interacts with VAP, the data presented is minimal and most experiments have next to no controls.

To further support our claims, we included co-immunoprecipitation data that shows VPS13A interacts with both VAP-A and VAP-B in an FFAT dependent manner (Figure 4F, supplementary Figure 4C). FFAT-dependent interaction between VPS13A and VAP-A was also demonstrated by Kumar et al., using immunolocalization studies overexpressing constructs harboring mutations in the FFAT domain and by using VAP-A-KO cells and investigate the localization of VPS13A overexpression in these cells.

4) The authors claim the protein's localization can switch to LDs. How these two different localizatios are achieved and or regulated is unclear.

We have now modified the manuscript and mention that there are two VPS13A localization patterns: VPS13A localizes at ER-Mitochondria contact sites and at the lipid droplets. Whether and how these two patterns interact is not clear. Under control conditions there are only a few lipid droplets present in the cells, so the lipid droplet vesicular pattern is only observed in some cells and a few VPS13A-positive circular structures are visible. When cells contain more lipid droplets, VPS13A is decorating the majority of them. Based on our data we cannot conclude whether this enrichment of VPS13A at lipid droplets is due to protein relocation or to a de novo synthesis. We can conclude that there is an enrichment of VPS13A associated with lipid droplets because there are more lipid droplets. This is confirmed by the fractionation studies (Figure 7) demonstrating that the lipid droplet fraction is enriched in VPS13A protein after addition of oleic acid (OA). The mechanism behind this enrichment could be explained by the presence of adaptor proteins at the surface of lipid droplets which compete for binding to VPS13A. Organelle membrane specific VPS13 adaptors competing for VPS13 binding have been found in yeast (Bean et., 2018). It could therefore be possible that human lipid droplets contain a VPS13A specific adaptor protein recruiting VPS13A to lipid droplets when they have been formed. We now combine all this information in our new model explained in the discussion and presented in Figure 10.

5) The authors claim VPS13A is important for LD homeostasis. However, the phenotype observed is mild at best and the characterization lacks even the most basic analyses (e.g. lipid content of the cells, localization of other LD proteins). In addition some of the measurements have problems (some of the values for LD sizes seem to be below the resolution of the light microscope). Moreover, in these experiments, the other Vps13 isoforms are not considered; overall this is a very preliminary analyses, and if anything, suggests a minor role in LD function at best.

To show differences in LD numbers, we have now replaced the data with LD number quantifications from VPS13A KO cells and their parental control cells. We also added data obtained in *Drosophila* eyes as suggested by reviewer 1, a phenotype rescued by overexpression of VPS13A in the mutant background. The increase in lipid droplets in human VPS13A KO cells is consistent with the increase in the *Drosophila* Vps13 mutant. We have removed the lipid droplet size measurements. We changed our text in such a way that we do not claim differences in lipid droplet homeostasis/metabolism. We do conclude that the number of lipid droplets is increased and that lipid droplet motility is negatively influenced by VPS13A. We discuss that the increase in lipid droplet numbers could be explained by decreased lipid droplet turnover.

6) The phenotype in Drosophila is interesting but also not comprehensively analyzed; only some EMs are shown. The authors would have to provide at least TG and other lipid measurements, and some evidence (e.g. by IF and light microscopy) that the structures observed are LDs.

We now have removed the EM data and have included instead IF and light microscopy analysis of *Drosophila* mutant eye in which lipid droplets are visualized using Nile red (Figure 9), as suggested by reviewer 1.

Reviewer #3:

Yeshaw and colleagues have explored the potential role of VPS13A a member of a small family of 4 related proteins and which is mutated in the neurological disorder Chorea Acanthocytosis. The gene encodes a very large cytosolic/membrane associated protein with conserved Chorein, DUF and ATG domains. Like for other family members, little is known about the exact role of VPS13A. Most knowledge is gathered through KD or KO strategies in different model systems, leading to the conclusion that VPS13A has a multitude of cellular functions. The authors explored here in more detail the localization and subcellular functional dynamics in cellular models. They demonstrate that VPS13A is involved in (and localizes to) membrane contacts between the ER and mitochondria; this interaction is calcium dependent and mediated through interaction of the FFAT domain with VAP-A, which they explored using mutational analysis. They further show that upon fatty acid addition, VPS13A is released from the ER and associates, in a FFAT-domain independent way, with lipid droplets. Functionally, VPS13A appears to affect LD size and its association with LDs slows their motility. The effects of VPS3 deficiency on LD appearance were finally validated in a mutant fly model using EM. Overall, the cell and molecular biology is of high quality and the authors make a major contribution to the potential role(s) of this VPS13 member, albeit the full mechanism and physiological importance with respect to the clinical phenotype in ChAc is not strongly revealed. A major concern remains the validation of the cellular data in mutant flies. The fly has only one VPS13 orthologue while in mammals, four variants are known, all with different functions (for instance the VPS13C has opposite effects on LDs compared to VPS13A) and links to very distinct diseases. It follows that the phenotype of the mutant fly cannot be correlated to the function of a specific variant, while the authors actually do that. A possible way to address this is to rescue the mutant fly with each human VPS13 variant and test to what extent they all rescue the phenotype or only certain features. In the best case, the common function among all VPS13 forms could be identified in this way. After all, to what extent is the observed cellular phenotype (smaller, more mobile LDs) correlated with the mutant fly phenotype (larger LDs)?

As explained above, we now provide explanations for the observed lipid droplet phenotypes in the discussion and discuss how this could link to the VPS13A-associated disease. For this we use our added data in combination with the two recent VPS13-related manuscripts in JCB.

As suggested by reviewer 1 we have now analyzed *Drosophila* eyes and showed that *Drosophila* Vps13 mutants have increased amount of lipid droplets in pigment cells of the eye. This phenotype is rescued by overexpression of human VPS13A in the *Drosophila* mutant background, suggesting a conserved function of human VPS13A in regulation of numbers of lipid droplets. The fly data are consistent with the phenotype observed in the now added analysis of VPS13A KO cells which also show an increase in lipid droplet numbers. In flies, we did not analyze the motility of lipid droplets compared to control because hardly any lipid droplets were observed in *Drosophila* wild type eyes. The decreased lipid droplet motility when decorated with VPS13A in the human cells as presented in Figure 8E-H can be explained by the results recently obtained in Kumar et al., 2018, showing that the VPS13A signal and lipid droplets are in close association with the ER, it could be possible as we now propose in our model that the reduced motility is because there is more attachment to the ER. This explanation was already proposed by reviewer 1 and in the Kumar et al. manuscript evidence for this is presented.

Specific comments that should be addressed:Figure 1C-D: the authors explore the membrane association of VPS13A using chemical agents. It surprises me why they don't use 'golden standard' approaches using bicarbonate, or detergents like TX114-phase partitioning or other that more selectively extract associated proteins.

For our membrane association studies we used slightly modified versions of published approaches (Holden and Horton, 2009; Mattie et al., 2018, Sugiura et al., 2017 and Vonk et al., 2017). We have now made this clearer in Materials and methods. We also include data that shows that VPS13A can be extracted with bicarbonate or can be cleaved with proteases from the mitochondrial surface (Figure 3—figure supplement 1).

Figure 2 and Figure 2—figure supplement 1: Truncated forms of VPS13A are used to show that the C-terminus is required for mitochondrial association. The authors refer to Figure 2—figure supplement 1 to state that the C-terminal region localized to the mitochondria in different cell lines: this is an overassumption as Figure 2D and Figure 2—figure supplement 1 only shows a co-staining of an ER marker with truncated VPS13A, not with a mitochondrial marker. Also the localization of the C-terminus is very different between different cell lines (compare A with C for instance: can the authors exclude that in A-B these structures are not other compartments like LDs?). The only evidence is in fact Figure 2B and 2E (triple staining). Small remark: the yellow arrow is not on the right position in panel A, merged inset.

Figure 2D is now replaced with new data to show the localization of VPS13A fragments to the mitochondria using the various constructs in combination with a mitochondrial marker in U2OS cells for optimal morphology (Figure 2E of the revised version). In addition, line scan co-localization analysis is also included for each set of GFP-VPS13A fragment and mitochondria.

We apologize for the confusion regarding the yellow arrow, this arrow is to indicate the direction and the position of the line scan, this is now indicated more clearly.

In different cell lines the morphology of the mitochondria is different, explaining the different localization patterns of the C-terminus construct in our original Figure 3—figure supplement 3. We agree that this is confusing and we now explain this better in the main text and in Figure 3—figure supplement 3B.

Figure 4E: same remark. Since only a co-staining with an ER-marker, VAP-A, is shown, one cannot conclude from these panels that VPS13AdeltaFFAT is shifted from ER to mitochondrial localization. Given that VPS13A can also locate to LDs, a triple staining or dual staining with a mitotracker and higher resolution (zoomed insets) is needed.

We like to note that mitochondria in HEK293 cells are not as filamentous as in other cell lines. In addition, VPS13A localization to lipid droplets is usually distinctively round (Video 1) unlike what is depicted in our original Figure 4E. We agree that this is confusing and we now explain this more clearly in the main text. We also have included new immunoprecipitation experiments that support the observation that VPS13AdeltaFFAT does not associate with the ER protein VAP-A. In addition, we now have added the requested localization studies comparing the expression of full length VPS13A with VPS13AdeltaFFAT in combination with a mitochondrial marker. This information is provided in Figure 4E and Figure 4—figure supplement 4B.

Figure 5C: these data show the relocation of GFP-VPS13A to LDs upon addition of OA. However these seem to be snapshots at different time points from different regions of the coverslip. Hence, the last sentence of this paragraph, 'Live-cell imaging showed that newly formed LDs… gradually acquired VPS13A…', is an overstatement as not the same LD is followed in time. Furthermore, one cannot deduce from these panels nor from the video that GFP-VPS13A goes to newly formed LDs, but instead associates/dissociates from existing LDs. Data in Figure 7E and Video 3 are also not really supporting this. To experimentally test this, probes like LiveDrop should be used as these mark the earliest stages of LD formation from the ER. Furthermore, is GFP-VPS13A recruited to LDs from nearby ER-mito contacts as this would suggest a more intricate association of ER-mito contacts with LD formation?

We have rephrased some statements and conclude that, under conditions of OA, VPS13A is enriched in the fraction containing lipid droplets; please see also our answer to reviewer 2 (for your convenience our answer is provided again below).

We have now modified the manuscript and mention that there are two VPS13A localization patterns: VPS13A localizes at ER-Mitochondria contact sites and at the lipid droplets. Whether and how these two patterns interact is not clear. Under control conditions there are only a few lipid droplets present in the cells, so the lipid droplet vesicular pattern is only observed in some cells and a few VPS13A-positive circular structures are visible. When cells contain more lipid droplets, VPS13A is decorating the majority of them. Based on our data we cannot conclude whether this enrichment of VPS13A at lipid droplets is due to protein relocation or to a de novo synthesis. We can conclude that there is an enrichment of VPS13A associated with lipid droplets because there are more lipid droplets. This is confirmed by the fractionation studies (Figure 7, Figure 7—figure supplement 2) demonstrating that the lipid droplet fraction is enriched in VPS13A protein after addition of oleic acid (OA). The mechanism behind this enrichment could be explained by the presence of adaptor proteins at the surface of lipid droplets which compete for binding to VPS13A. Organelle membrane specific VPS13 adaptors competing for VPS13 binding have been found in yeast (Bean et al., 2018). It could therefore be possible that human lipid droplets contain a VPS13A specific adaptor protein recruiting VPS13A to lipid droplets when they have been formed. We now combine all this information in our new model explained in the discussion and presented in Figure 10. (see also our answer to reviewer 1)

Figure 6: Using cell fractionation, the authors show that OA induces a shift of VPS13A to the floating LD fraction. The quantification shows clearly a trend but are these increases statistically significant (B-B' and D-D')? This is important since VAP-A also increases indicating at least some contamination? Some antibodies also give multiple bands: specific bands to PLIN2 and LAMP1 should be more clearly indicated to interpret the data. The authors have also used two concentrations of OA (250 and 500µM). While panel 6F shows the dose-dependent increase in VPS13A with LDs (using imaging), this is less obvious for the biochemical (flotation gradients) data. In fact, there is little difference when looking at the relative shifts of VPS13A and PLIN2 between 250 and 500µM.

*VAP was also found to be present in lipid droplet fractions in proteomic studies (Cermelli et al., 2006),* nonetheless, it is unclear whether this is a contamination or not. The enrichment of PLIN2 is indicative that indeed lipid droplet fractions were obtained, together with the knowledge that lipid droplets are present in the floating and lighter fractions and that lipid droplet numbers are strongly decreased under starvation conditions. We have indicated the specific PLIN2 band with an asterisk. The LAMP1 signal obtained with the antibody is visible as a smear in the higher fractions, but not in the lighter fractions; we do not have an explanation for this, it could be explained by post translational modifications of LAMP1.

Figure 7: In some cases, like in this figure the authors switch to U2OS cells. Why? I would also include a zoomed inset for panel D: the shifts in frame 1 vs frame 2 are difficult to discern. Also these images do not really allow to distinguish large from small LDs (they seem in both wt and siRNA to be overall small). Better images are needed that clearly demonstrate the point the authors are making. In addition, instead of measuring% overlap, it might be better to measure 'distance travelled' as a readout for mobility (and thus to show differences). The conclusion of this paragraph is also overstated: 'In the absence of VPS13A' should be 'When VPS13A is downregulated'. Did the authors generate CRISPR/Cas9 KO cells of VPS13A and if so, are the phenotypes worse?

The data in Figure 7A-C is now replaced with data obtained in VPS13A KO cells and presented in Figure 5 and Figure 8.

We like to note that we did not include the quantification of large lipid droplets versus small lipid droplets in the revised version, we focused now on the amount of lipid droplets in VPS13A depleted and VPS13A containing cells cultured under control conditions. The numbers of lipid droplets are increased in control culturing conditions in the newly generated VPS13A KO cells compared to the parental line. The phenotype is indeed stronger as compared to the results obtained with RNAi. An increase in the amount of lipid droplets in VPS13A KO cells is consistent with data obtained in the *Drosophila* Vps13 mutant eye (Figure 9). We provide an additional assay (Figure 8G and H) to visualize how VPS13A is influencing motility and a different read-out was used.

In some of the panels with confocal data, the images seem to be moved or shifted. For instance in Figure 1—figure supplement 1C, the Rab5QL enlarged endosomes have a suspicious double membrane and the Rab7QL endosomes look fuzzy. Another example is Figure 7A where each dot is a little stripe as this is a timelapse (or like a picture of the night sky with long opening time).

We have now adjusted these figures and provide better quality images.

In the Discussion section the authors refer to the work of the Bellen lab to state that LDs are formed in glia cells in response to oxidative stress. In the meantime, they published a follow-up paper on a non-cell autonomous mechanism which should be included (Liu et al., 2017) and may help to explain (the differences with) the observed phenotype.

*We have adjusted the discussion dramatically, we included the Liu et al. Cell Met manuscript and believe* a clearer picture is arising about what the role of VPS13A is and how VPS13A-associated neurodegeneration could arise.

[Editors' note: the author responses to the re-review follow.]

The manuscript has been improved but there are some remaining issues that need to be addressed before acceptance, as outlined below:Overall, the reviewers are satisfied and they note the value of the paper, also in the light of currently emerging publications on Vps13. However, they agree that some of the aspects still need strengthening. Specifically and most importantly, the reviewers are not yet fully convinced by the data concerning a putative switch from ER-mito to ER-LDs. There is a general agreement that this switch is not sufficiently substantiated. The authors could still address this with their current data by quantifications and providing controls.

We agree that a putative switch from ER-mito to lipid droplets is not sufficiently substantiated. Therefore in contrast to our initial manuscript we do not claim this in the revised version, we discuss this possibility instead. We have now made more clear in the discussion that more experiments would be required to demonstrate a possible relocation of VPS13A from ER-mito to lipid droplets. We believe that quantification of our current data may not be enough to convincingly demonstrate this.

Furthermore, in the light of recent publications please make sure to avoid the statements concerning an "unknown function" of Vps13, such as the one in the Introduction.

We have changed this accordingly.